# Sub-4 nanometer porous membrane enables highly efficient electrodialytic fractionation of dyes and inorganic salts

Jiuyang Lin [1,2,3], Zijian Yu[3], Tianci Chen[2], Junming Huang[3], Lianxin Chen[3], Jiangjing Li[3], Xuewei Li[2], Xiaolei Huang [2], Jianquan Luo [4], Elisa Yun Mei Ang [5], William Toh[6], Peng Cheng Wang [5], Teng Yong Ng[6], Dong Han Seo[7], Shuaifei Zhao [8], Kuo Zhong [9] ✉, Ming Xie [10] ✉, Wenyuan Ye [1] ✉, Bart Van der Bruggen [11,12] & Yinhua Wan[2] ✉

During the synthesis of dyes, desalination of high-salinity dye-containing waste liquor is a critical premise for high-quality, clean dye production. Conventional membrane processes, such as electrodialysis, nanofiltration and ultrafiltration, are inevitably subjected to serious membrane fouling, deteriorating the dye/salt fractionation efficacy. Integrating the technical merits of electrodialysis and pressure-driven membrane separation, we devise an electro-driven filtration process using a tight ultrafiltration membrane as alternative to conventional anion exchange membrane for rapid anion transfer, in view of dye desalination and purification. By employing a sub-4 nanometer tight ultrafiltration membrane as anion conducting membrane, the electro-driven filtration process achieves 98.15% desalination efficiency and 99.66% dye recovery for one-step fractionation of reactive dye and NaCl salt, markedly outperforming the system using commercial anion exchange membranes. Notably, the electro-driven filtration system displays a consistently high and stable fractionation performance for dyes and salts with unprecedentedly low membrane fouling through an eight-cycle continuous operation. Our results demonstrate that the electro-driven filtration process using nanoporous membranes as high-performance anion conducting membranes shows a critical potential in fractionation of organic dyes and inorganic salts, unlocking the proof of concept of nanoporous membranes in electro-driven application.

In the manufacturing sector for target organics in industries, such as food, pesticide, pharmaceutical, and dye, abundant inorganic salts are generated, thus minimizing the purity of the target organics[1-6]. Specifically, the textile industry, as the pillar industry in most of the low/middle-income countries, consumes a large quantity of synthetic dyes for high-quality fabric production[7]. Around 1 million tons of synthetic dyes are produced annually for dyeing purposes[8]. During the dye synthesis process, acid and alkali would be subsequently used for neutralization, leading to the generation of inorganic salts (mainly NaCl or $Na_2SO_4$) as by-products[9,10]; on the other hand, the inorganic salts were also widely added as the sedimentation agent to precipitate the dyes, thereby achieving rapid extraction of raw dye products[11]. Unfortunately, the extracted dyes contain considerable amounts of inorganic salts, which deteriorate the purity of the raw dye products[2,12].

Therefore, the resultant dye products with low purity are inappropriate to be applied in digital inkjet printing applications, which fails to drive technological innovation of the textile industry. Generally, a sophisticated solvent extraction process is employed to purify the dyes[13,14], which not only leads to a marked increase in the cost of purified dye products for limiting their application in digital inkjet printing but also causes secondary pollution and insufficient recovery of inorganic salts as a useful resource. From the perspective of circular economy and towards net-zero discharge, it is necessary to efficiently separate the dyes and salts from raw high-salinity dye-containing liquor for simultaneous dye purification and salt recovery[15,16].

Membrane separation processes provide a technology platform to fractionate the organic compounds and inorganic salts, given their impressive selectivity[17,18]. For example, electrodialysis is proposed to desalinate high-salinity dye-containing liquor, which enables the directed transfer of anion and cation through anion exchange membranes and cation exchange membranes under a direct current field[19]. However, dyes with negative charges would preferentially accumulate onto the surface or pore structure of anion exchange membranes (with positive charges) via the electrostatic attraction effect, inducing severe membrane fouling[19,20]. On the other hand, the positive charge density of the fouled anion exchange membranes would be markedly diminished via charge neutralization with negatively charged dyes[21]. These two adverse effects would increase the electric resistance of anion exchange membranes, tremendously jeopardizing anion transfer for insufficient desalination efficiency.

Apart from electrodialysis, pressure-driven membrane processes, such as nanofiltration, have been established as an effective approach for the fractionation of dyes and inorganic salts[22,23]. Based on the synergistic effects of size exclusion and electrostatic repulsion, nanofiltration membranes can effectively retain the organics with molecular weights of 200–2000 Da[24–26]. The widely-used commercial nanofiltration membranes have a narrow pore size distribution (i.e., 0.5–2.0 nm), allowing for a strong electrostatic repulsion for partial transmission of inorganic salts and thus leading to a moderate selectivity between dyes and inorganic salts[27]. Furthermore, cake-enhanced concentration polarization[28] and dye-induced membrane fouling[29,30], which inevitably occur during the pressure-driven nanofiltration procedure, initiate a drastic reduction in membrane permeation flux, thus minimizing the fractionation efficacy between dyes and inorganic salts.

Tight ultrafiltration offers a conceptual approach for the desalination of high-salinity dye-containing waste liquor[31,32]. Interestingly, dye aggregation occurs between the adjacent benzene ring groups of dyes via hydrophobic interactions, leading to the formation of dye clusters with larger size[33–36]. Therefore, the implementation of tight ultrafiltration membranes with molecular weight cutoff of 2000–5000 Da (i.e., pore size of 2–5 nm) features an acceptably high dye rejection based on the enhanced size exclusion effect against dyes[37]. Moreover, the nearly complete passage of inorganic salts can be obtained by tight ultrafiltration membranes due to diminished electrostatic repulsion against ions caused by their large pore size, enhancing the selectivity between dyes and inorganic salts during their fractionation. Nevertheless, the flux decline of tight ultrafiltration membranes caused by the dye cake layer and pore blockage still pose a tough challenge. Furthermore, diafiltration with tight ultrafiltration-based procedure is ineluctably utilized to fractionate the dyes and inorganic salts for dye desalination via consumption of pure water with high diavolumes, potentially giving rise to a loss of valuable targeted dyes for reduced productivity[38].

Herein, we designed an electro-driven ultrafiltration system, through combining the technical advantages of electrodialysis and pressure-driven ultrafiltration (i.e., tight ultrafiltration). Specifically, we employed a sub-4 nanometer porous polyethersulfone membrane (molecular weight cutoff of 2292 Da) in a conventional

electrodialysis stack as an anion conducting membrane, which acts as an alternative to an anion exchange membrane (Fig. 1A). On the one hand, with the aid of the nanoporous structure (mean effective pore size of 1.52 nm) of the membrane, anions (e.g., Cl⁻ ions) can transport from the feed chamber through the membrane pores and subsequently migrate to the concentrate chamber under a direct current field for desalination. On the other hand, the size exclusion effect of the sub-4 nanometer porous membrane can effectively retain the dyes, establishing an impressive selectivity between reactive dye and NaCl for achieving their one-step, efficient, and viable fractionation, as illustrated by molecular dynamics simulation. In particular, the sub-4 nanometer porous membrane yielded a 98.15% desalination efficiency with 99.66% dye recovery during electro-driven fractionation of the reactive dye/NaCl mixture solution. Furthermore, the sub-4 nanometer porous membrane exhibited a consistently steady dye/salt fractionation with ignorable fouling propensity during the eight-cycle electro-driven filtration process under the direct current field, which markedly surpassed the commercial anion exchange membrane. Our study demonstrates the proof-of-concept for this electro-driven membrane-based filtration system and its associated mass transfer behaviors, paving a facile pathway to one-step fractionation of organic dyes and inorganic salts for desalination and purification of high-salinity dye-containing liquor.

## Results
### Characterization of nanoporous membrane
The morphology of the membrane is critical for solute rejection in both pressure-driven and electro-driven filtration. The tested membrane had no visible pores on the surface, as observed by scanning electronic microscopy (SEM) (Fig. 1B). In particular, the sub-4 nm pore size of the tested membrane represented a tight nanoscale surface structure. Furthermore, the tested membrane showed a skin layer with a thickness of ~500 nm (Fig. 1C), which provided a molecular weight cut-off (MWCO) of 2292 Da with mean effective pore diameter of 1.52 nm (Fig. 1D), as measured from the rejection of different molecular-weight polyethylene glycol polymers. The cross-section SEM image illustrated that the sub-4 nanometer porous membrane established a hierarchical porous, spongy structure, which would offer a strong mechanical support during both pressure-driven and electro-driven filtration process. Furthermore, the Fourier transform infrared spectroscopy (FTIR) measurement of the tested membrane shows the intrinsic peaks at the wavenumbers of 1576.5/1484.4/1411.1, 1321.0/1296.6/1147.4/1103.6, 1236.6 and 717.4 cm⁻¹, which are associated with the stretching vibration of aromatic ring groups (benzene rings), sulfone groups, aromatic ether structure, and C-S-C group, respectively (Supplementary Fig. 1A)[39,40]. The characteristic stretching vibrations demonstrated that the tested membrane is the polyethersulfone-based membrane, which can be further confirmed by the X-ray photoelectron spectroscopy (XPS) scan spectra of C 1 s, O 1 s and S 2 p (Supplementary Fig. 1B)[41,42].

Zeta potential measurements revealed that the tested membrane displayed a negatively charged property over the tested pH range from 4.1 to 9.7. Particularly, a linear negative correlation between the zeta potential and pH can be clearly observed (Fig. 1E). On the other hand, the membrane offered a moderately hydrophilic surface, as proved by the dynamic water contact angle measurement (Fig. 1F). Such a hydrophilic surface can allow for solution wetting, imparting this sub-4 nanometer porous membrane with a low specific areal electric resistance (i.e., $9.32 \pm 0.08\,\Omega\,cm^2$) for anion transfer under a direct current field.

### Pressure-driven filtration performance
To elucidate the selectivity between dye and NaCl of the tested sub-4 nanometer porous membrane, the pressure-driven membrane

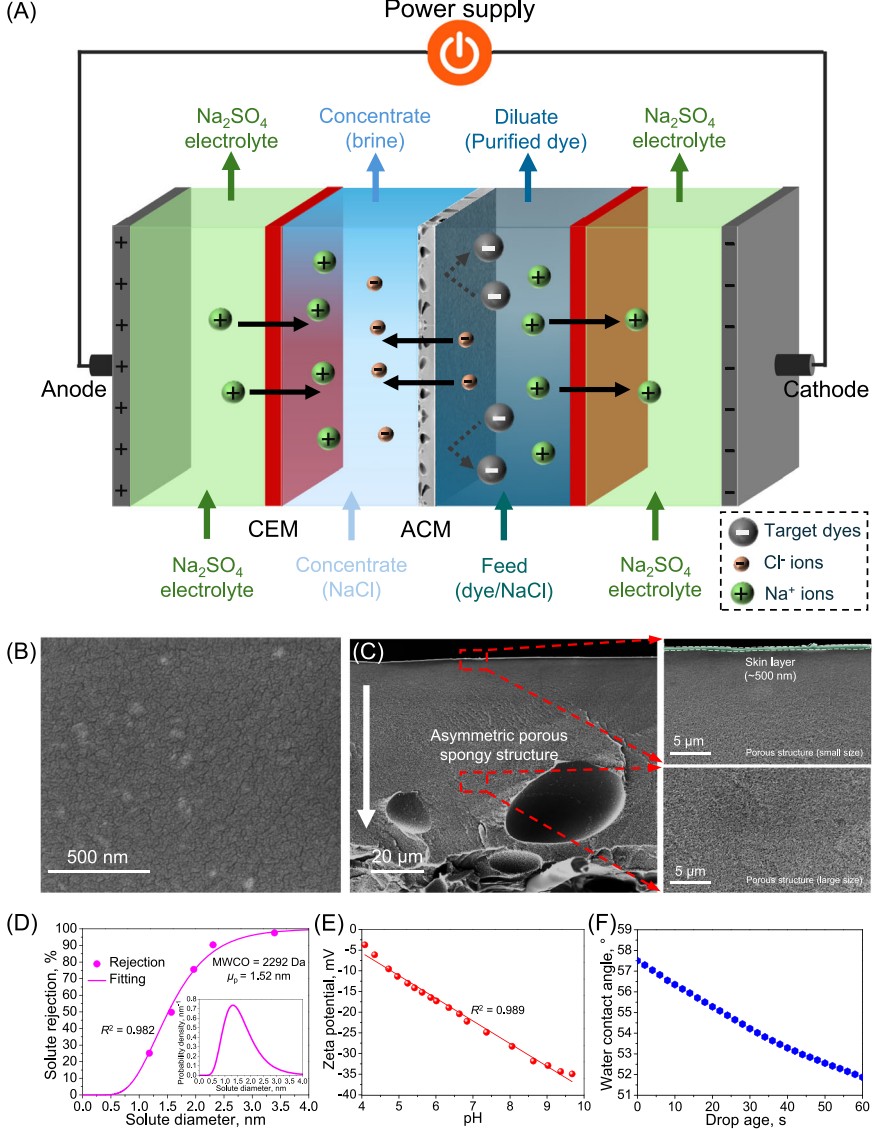

**Fig. 1 | Design of an electro-driven filtration system using a sub-4 nanometer porous membrane as anion conducting membrane for dye/NaCl fractionation. A** stack configuration of the electro-driven filtration process. **B** Surface SEM image of the sub-4 nanometer porous membrane. **C** Cross-sectional SEM images of the sub-4 nanometer porous membrane. The cross-section SEM image shows increasing pore size from the top to the bottom of the membrane, indicating a hierarchical porous spongy structure. **D** Molecular weight cutoff and pore size distribution of the membrane (inset: membrane pore size distribution). **E** Zeta potential of the membrane. **F** Membrane hydrophilicity. CEM: Cation exchange membrane; ACM: Anion conducting membrane (sub-4 nanometer porous membrane).

separation behavior in various solutions (i.e., dye or NaCl solution) was explored (Supplementary Figs. 2–4 and Fig. 2).

Initially, the tested sub-4 nanoporous membrane experienced a minimal salt rejection in the NaCl solutions since the large pore size of the tested sub-4 nanometer porous membrane not only minimized the size exclusion effect but also undermined the electrostatic repulsion effect between the membrane and ions, which facilitates the salt transmission[43]. Specifically, the salt rejection of the sub-4 nanometer porous membrane in a 0.5 g L$^{-1}$ NaCl solution was as low as 16.71 ± 0.93% (Supplementary Fig. 2). As the NaCl concentration increased to 14.6 g L$^{-1}$, the membrane yielded a nearly complete salt transmission (i.e., 1.33 ± 0.18% NaCl rejection), due to the fact that the elevated salt concentration can suppress the Debye screening length of the sub-4 nanometer porous membrane, intensifying the electrostatic shielding against charged ions for reducing the salt rejection[44,45].

On the other hand, the sub-4 nm porous membrane yielded an expectedly high rejection (> 99.4%) against reactive dyes with different molecular weights (ranging from 626.5 to 1338.1 g mol$^{-1}$)

(Supplementary Fig. 3 and Supplementary Fig. 4), highlighting a promising potential of this nanoporous membrane for screening the reactive dyes. Such an impressive rejection of reactive dyes is mainly due to the aggregation of reactive dyes for the formation of dye clusters with enlarged size via hydrophobic interaction between the benzene ring structure of adjacent reactive dye molecules (Supplementary Fig. 5 and Supplementary Movie 1)[35,46]. Therefore, the dye clusters can be easily retained by the sub-4 nanometer porous membrane through enhanced sized exclusion, which also can be reflected by extremely high dye rejection in the molecular dynamics simulation (Supplementary Fig. 6 and Supplementary Movie 2).

Generally, a dye-containing waste liquor with high salinity can be generated during the synthesis of reactive dyes. The co-existence of NaCl with dyes could potentially affect the filtration behavior of the sub-4 nm membrane (Fig. 2). For example, when the NaCl concentration in the reactive black 5/NaCl mixture solution increased from 0 to 14.6 g L$^{-1}$, the corresponding rejection of reactive black 5 decreased (Fig. 2A), likely due to the intensification in electrostatic shielding

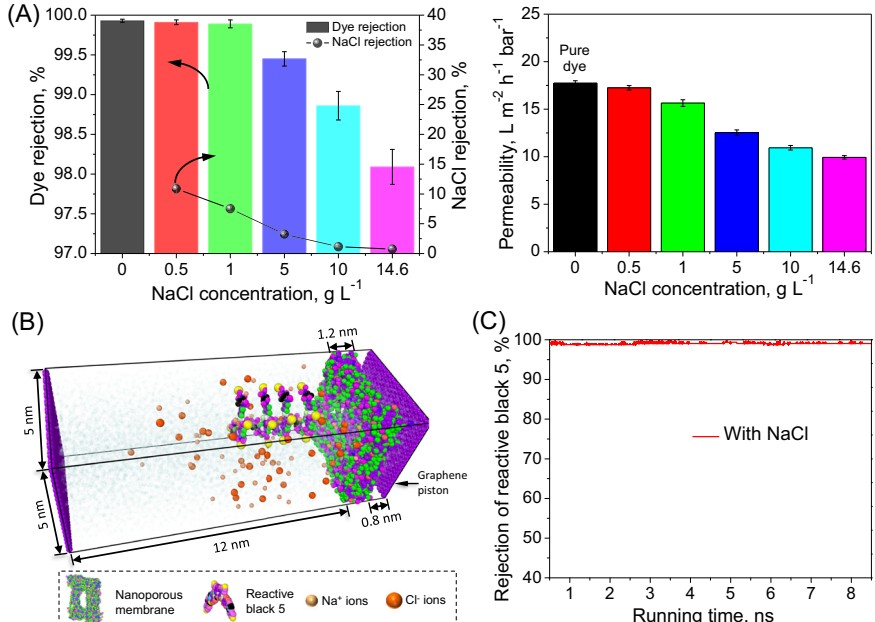

**Fig. 2 | Filtration performance of the sub-4 nanometer porous membrane in reactive black 5/NaCl mixture solution with different salinities. A** Rejection of solutes (reactive black 5 and NaCl) and permeability; error bars in the figure represent the standard deviation of three measurements. **B** Details of the molecular dynamics simulation domain. **C** Rejection of reactive black 5 by molecular dynamics simulation.

effect caused by the elevated salt concentration, diminishing the electrostatic repulsion against the dyes[38]. Simultaneously, the membrane swelling caused by the elevated salt concentration, inducing a slight increase in the pore size of the membrane surface[47], could potentially facilitate the transfer of the dye through the sub-4 nanometer porous membrane. On the other hand, the formation of dye clusters via hydrophobic interaction of dye molecules can partially offset the negative effect of elevated NaCl concentration on the dye rejection, yielding a slight decline in dye rejection from $99.93 \pm 0.02\%$ to $98.09 \pm 0.22\%$ with increasing salt concentration (Fig. 2A). This can be demonstrated by the consistently high dye rejection in the molecular dynamics simulation during the filtration of dye/NaCl mixture solution (Fig. 2B, Fig. 2C and Supplementary Movie 3). However, the selectivity between NaCl and reactive black 5 was declined drastically from 990.2 to 52.0 with increasing NaCl concentration (Supplementary Fig. 7), mainly due to the fact that the electrostatic shielding effect on the membrane surface was amplified with increasing NaCl concentration, which undermines the electrostatic repulsion with reduced dye retention, resulting in dye loss (i.e., 1.71%) through the membrane permeate and thus compromising the dye/salt fractionation during the pressure-driven constant-volume diafiltration process for dye purification (Fig. 3 and Supplementary Table 1). On the other hand, the permeability of the membrane also showed a significant decrease with increasing NaCl concentration (up to $14.6\,g\,L^{-1}$) by 44.0%, declining from $17.73 \pm 0.26\,L\,m^{-2}\,L^{-1}\,bar^{-1}$ to $9.92 \pm 0.19\,L\,m^{-2}\,L^{-1}\,bar^{-1}$ (Fig. 2A), which can markedly deteriorate the dye/salt fractionation efficacy through pressure-driven filtration.

## Electro-driven filtration performance

Alternatively, the tested sub-4 nanometer porous membrane was employed as an anion conducting membrane in the electro-driven filtration process to assess its anion transfer capacity (Fig. 4). During the electro-driven filtration, a linear decay in the conductivity of the pure NaCl solutions as the feed (diluate) was observed (Fig. 4A), indicating that $Na^+$ and $Cl^-$ ions in the feed consistently migrated through the cation exchange membrane and the sub-4 nanometer porous membrane, respectively, under the direct current field for desalination. Such

a phenomenon can be demonstrated by the opposite moving direction of $Cl^-$ and $Na^+$, as reflected through the positive drift velocity of $Cl^-$ ions and the negative drift velocity of $Na^+$ through molecular dynamics simulation (Supplementary Fig. 8 and Supplementary Movie 4). Particularly, the sub-4 nanometer porous membrane exhibited a 98.29%–98.43% desalination efficiency for the pure NaCl solution ($14.6\,g\,L^{-1}$) under a direct current field (Fig. 4B). On the other hand, a linear relationship between the anion (i.e., $Cl^-$) transfer rate and the applied current intensity can be obtained (Fig. 4C), suggesting that the anion transfer was dominated by the direct current field as driving force. Simultaneously, the salt concentration in the feed had no impact on the anion electrodialytic transfer of the sub-4 nanometer porous membrane for desalination, reflected by the consistent NaCl transfer rate (ca. $3.85\,g\,L^{-1}\,h^{-1}$) (Supplementary Fig. 9). The marked anion transfer under the direct current field was mainly due to extra nanochannels and short anion diffusion path (i.e., ~500 nm) of the inherently thin skin layer of the sub-4 nanometer porous membrane, which acted as a critical role of high-performance anion conducting membrane (Fig. 4D). Therefore, such a powerful anion conduction property of the sub-4 nanometer porous membrane offers an insightful and effective platform in anion electrodialytic transfer for sufficient desalination.

Subsequently, the electrodialytic fractionation efficacy of the sub-4 nanometer porous membrane for dye and salt in the reactive black 5/NaCl mixture solution was explored (Fig. 5). Fast electrodialytic transfer of $Cl^-$ ions endowed the sub-4 nanometer porous membrane with a 98.15% desalination efficiency in the reactive black 5/NaCl mixture solution (Fig. 5A and 5B). Simultaneously, only an extremely limited quantity of reactive black 5 (i.e., $3.4\,mg\,L^{-1}$) penetrated through the sub-4 nanometer porous membrane and thus migrated to the concentrate chamber (Fig. 5B), achieving an unparallelly high dye recovery (i.e., 99.66%) (Supplementary Table 1) for an adequate fractionation of reactive black 5 and NaCl. Thereby, the sub-4 nanometer porous membrane with a thin skin layer provided sufficient nanochannels for efficient transfer of $Cl^-$ ions, but strongly repelled reactive black 5 based on intensified size exclusion (Supplementary Movie 5), achieving an exceptionally high permselectivity between NaCl and reactive black 5 (i.e., 15561.7) for one-step dye/NaCl electrodialytic

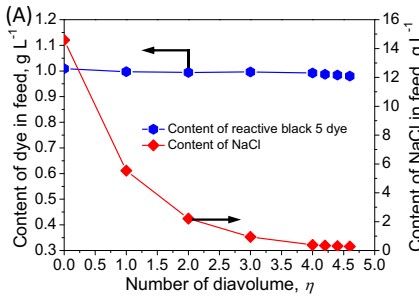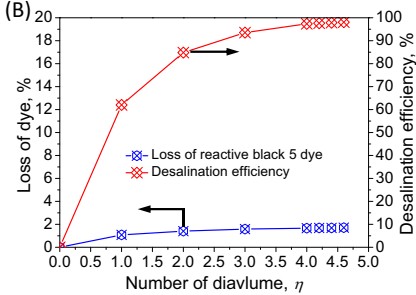

**Fig. 3 | Pressure-driven constant-volume diafiltration using the sub-4 nanometer porous membrane in the reactive black 5/NaCl mixture solution for fractionation of dye and NaCl. A** Content of reactive black 5 and NaCl in the feed at different diavolumes. **B** Loss of reactive black 5 and desalination efficiency.

fractionation (Supplementary Fig. 10). Moreover, such an electro-dialytic fractionation process, which was equipped with the sub-4 nanometer porous membrane as an anion conducting membrane, remarkably outperformed the pressure-driven constant-volume dia-filtration process using the sub-4 nanometer porous membrane as a tight ultrafiltration membrane (Fig. 3 and Supplementary Table 1) for reactive black 5/NaCl fractionation, in view of effective dye desalination.

To further demonstrate the fouling propensity of the sub-4 nanometer porous membrane as anion conducting membrane in long-term fractionation of dye and NaCl, an eight-cycle electrodialytic filtration operation was performed based on the sub-4 nanometer porous membrane using the reactive black 5/NaCl mixture solution (Fig. 6). Particularly, virtually identical fractionation for reactive black 5 and NaCl in each cycle operation during this eight-cycle electrodialytic filtration was yielded, which was demonstrated by the consistent decay in feed conductivity and increase in the concentrate conductivity (Fig. 6A). This continuous eight-cycle electrodialytic filtration featured a preponderant long-term stability of the sub-4 nanometer porous membrane for steadily exceptional desalination (desalination efficiency of 98.09%–98.22%) (Fig. 6B) and dye recovery (99.63%-99.69%) (Fig. 6C), demonstrating an impressive antifouling property, which can be reflected by the slight boost in specific areal membrane resistance after the eight-cycle electrodialytic filtration (Supplementary Fig. 11). Overall, such an outstanding antifouling performance is mainly ascribed to the fact that the sub-4 nanoporous membrane can efficiently repel the reactive dye clusters with larger sizes via enhanced steric exclusion effect, which pose no interference on anion transfer behavior (Fig. 6D).In addtion, the negative charge carried by the sub-4 nanometer porous membrane surface helped to strengthen the electrostatic repulsion against reactive black 5, partially mitigating the membrane fouling, for enhanced anion transfer in high-salinity dye-containing liquor.

## Discussion

The tested sub-4 nanometer membrane established an enhanced anion electrodialytic transport via its intrinsically nanoporous structure. Molecular dynamic simulations of anion transport were further conducted to illustrate the ion transfer mechanism of the sub-4 nanometer porous membrane as anion conducting membrane under a direct current field (Fig. 5C, D and Supplementary Movie 5). Due to the direct current field implementation across the cation exchange membrane and sub-4 nanometer porous membrane, Cl⁻ ions in the feed were forced to migrate through the nanochannels of the sub-4 nanometer porous membrane at a drift velocity of 2.7E-4 Å fs⁻¹ (Fig. 5D); on the other hand, Na⁺ ions in the feed were forced to migrate through the cation exchange membrane at the opposite direction for eletrodialytic desalination of the reactive black 5/NaCl mixture solution. In the contrary, the movement of the reactive black 5 is less affected by the applied direct current field. Specifically, reactive black 5 molecules, with larger size, were observed to have an extremely low drift velocity

(i.e., 1.1E-5 Å fs⁻¹) for moving towards the sub-4 nanometer porous membrane (Fig. 5D), which is much less than Cl⁻ ions. Therefore, based on the large difference in drift velocity between reactive black 5 and NaCl, the directed transfer status for both reactive black 5 and Cl⁻ allows for their selective fractionation. Moreover, with the aid of the enhanced size exclusion effect, reactive black 5 was sufficiently intercepted by the sub-4 nanometer porous membrane for effective one-step fractionation of the reactive dye and Cl⁻ ions (Supplementary Movie 5).

Generally, the conventional electrodialysis equipped with an anion exchange membrane offers a universal strategy for desalination and purification. However, the anion exchange membrane was subjected to severe fouling during a three-cycle electrodialytic filtration for fractionation of reactive black 5 and NaCl from their mixture solution (Supplementary Fig. 12). This was mainly ascribed to the strong electrostatic attraction effect between reactive black 5 with negative charges and anion exchange membrane carrying positively-charged quaternary ammonium groups (Supplementary Fig. 12C), which not only caused pore blocking of the anion exchange membrane, but also tremendously diminished its positive charge density. Such a physicochemical interaction between dye and anion exchange membrane can lead to a drastic increase in its specific areal resistance, in this case by a factor of 43.0, from 1.98 to 87.07 Ω cm² (Supplementary Fig. 12D). This remarkably limits its anion transport and compromises the desalination efficacy (Supplementary Fig. 12B). Particularly, the desalination efficiency of the commercial anion exchange membrane declined from 92.87% to 84.98% after the three-cycle electrodialytic filtration operation. Accordingly, the content of NaCl in the diluate solution maintained at the level of 1.04–2.19 g L⁻¹, demonstrating an insufficient desalination efficiency by conventional electrodialysis equipped with the commercial anion exchange membrane for the reactive dye/salt mixture solution. Simultaneously, the fouling of the anion exchange membrane caused by reactive dye tremendously extended the operating time from 300 min to 350 min during each filtration cycle for desalination after the three-cycle conventional electrodialysis (Supplementary Fig. 12A).

Therefore, the use of the sub-4 nanometer porous membrane as anion conducting membrane in the electrodialytic filtration process propelled an effective and one-step fractionation of reactive dyes and NaCl from high-salinity dye-containing liquor, markedly surpassing the commercial anion exchange membrane, in view of dye desalination and purification. This study sought to provide a proof-of-concept in performance demonstration of the use of nanoporous anion conducting membranes as an alternative to anion exchange membranes with high fouling propensity, in electrodialysis applications for efficient dye purification and salt reuse from high-salinity dye-containing liquor.

## Methods
### Materials and chemicals
The commercial sub-4 nanometer porous membrane was kindly supplied from Guangdong Yinachuan Environmental Technology (China)

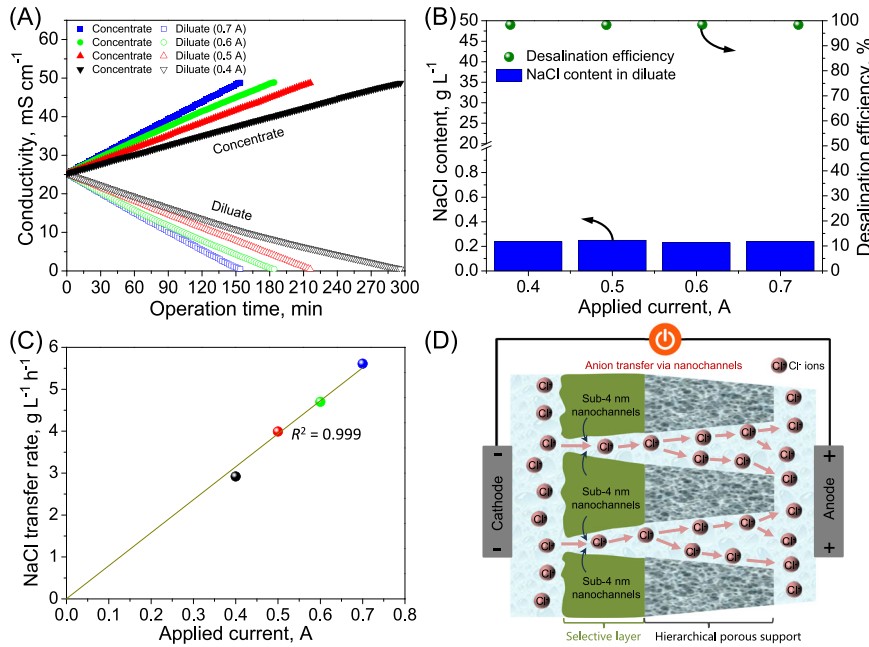

**Fig. 4 | Electrodialytic filtration performance of the sub-4 nanometer porous membrane as a role of anion conducting membrane in pure NaCl solution at different current intensities. A** Evolution of conductivity in both concentrate and diluate solutions. **B** NaCl concentration in the diluate and desalination efficiency. **C** Electro-driven transfer rate of Cl⁻ ions through the sub-4 nanometer porous membrane. **D** Demonstration of electrodialytic transport behavior of Cl⁻ ions through the sub-4 nanometer porous membrane.

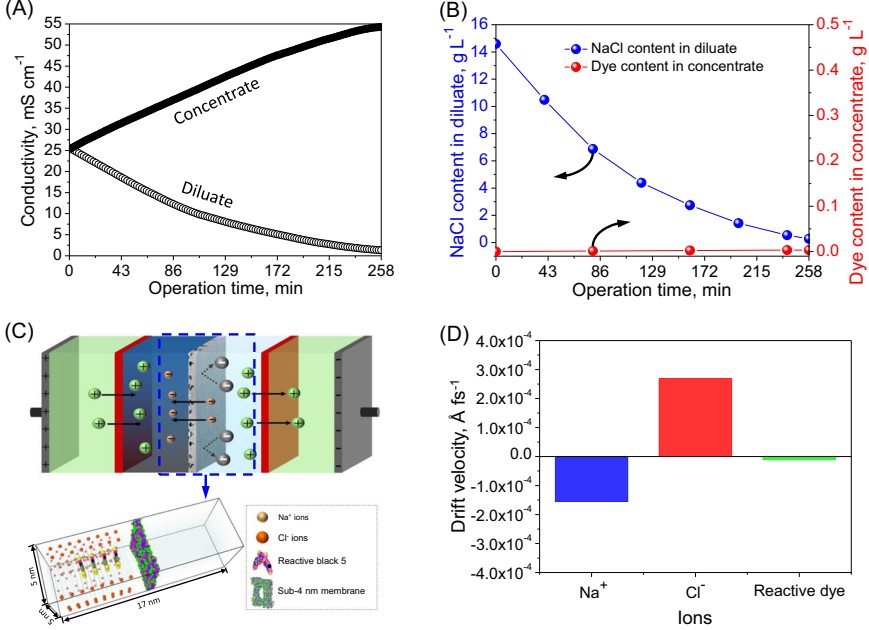

**Fig. 5 | Electrodialytic filtration performance of the sub-4 nanometer porous membrane as a role of anion conducting membrane in the reactive black 5/ NaCl mixture solution for fractionation of dye and NaCl. A** Evolution of conductivity in both the concentrate and diluate. **B** NaCl content in the diluate and dye content in the concentrate. **C** Details of the molecular dynamic simulation domain. **D** Drift velocity of Na⁺, Cl⁻, and reactive black 5 in the molecular dynamic simulation system.

and used as anion conducting membrane during electro-driven membrane-based filtration process. The commercial cation exchange membrane (TWEDC1S70) and anion exchange membrane (TWE-DA1R70) in the electrodialytic filtration process were purchased from Shandong Tianwei Membrane Technology Co., Ltd. (China).

Dyes with different molecular weights, including reactive blue 19 (molecular weight of 626.5 g mol⁻¹), reactive blue 4 (molecular weight of 637.4 g mol⁻¹; dye content of 35%), reactive red 24 (molecular weight of 788.1 g mol⁻¹; intensity of 100%), reactive blue 13 (molecular weight of 924.2 g mol⁻¹), reactive red 180 (molecular weight of 933.8 g mol⁻¹; purity of 95%), reactive black 5 (molecular weight of 991.8 g mol⁻¹; dye content of ≥ 50%), reactive red 195 (molecular weight of 1136.3 g mol⁻¹; intensity of 100%) and reactive red 120 (molecular weight of 1338.1 g mol⁻¹; dye content of ≥50%) were employed as the model dyes.

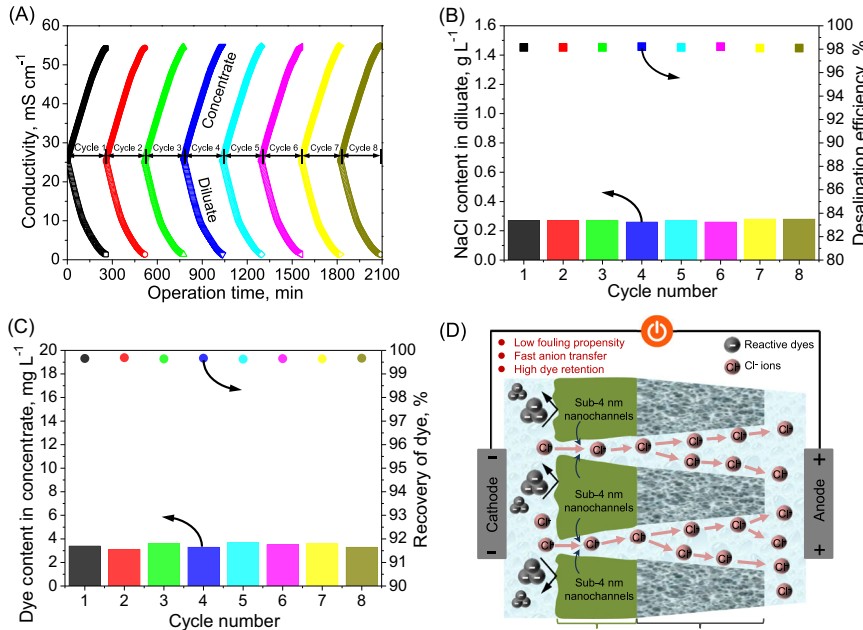

**Fig. 6 | Eight-cycle electrodialytic filtration using the sub-4 nanometer porous membrane as anion conducting membrane for fractionation of dye and NaCl from the reactive black 5/NaCl mixture solution. A** Evolution of conductivity in the concentrate and diluate. **B** NaCl content in the diluate and desalination efficiency. **C** Dye content in the concentrate and recovery rate. **D** Illustration of anion (Cl⁻ and reactive black 5) transfer of the sub-4 nanometer porous membrane for electrodialytic fractionation of the reactive black 5/NaCl mixture solution.

Reactive blue 19, reactive blue 13, reactive black 5, and reactive red 120 were purchased from Shanghai Aladdin Biochemical Technology Co., Ltd. (China), while reactive blue 4, reactive red 24, reactive red 180, and reactive red 195 were purchased from Macklin Chemical Co., Ltd. (China). Sodium chloride (NaCl) with a 99% purity was purchased from Sinopharm Chemical Reagent Co., Ltd. (China) to assess the filtration performance of the membrane. All the chemicals were used as received.

## Membrane characterization

The surface and cross-section morphology of the sub-4 nanometer porous membrane was observed by SEM (NOVA NanoSEM 230, FEI, USA). The chemical composition of the membrane was characterized by FTIR (Nicolet iS5, Thermo Fisher Scientific, USA) and XPS (Axis Supra +, Shimadzu, Japan) analysis. The surface wettability of the membrane in terms of dynamic water contact angle was determined through an optical surface analyzer (OSA200, Ningbo NB Scientific Instruments Co., Ltd., China). The surface charge of the membrane in terms of zeta potential was measured using an electrokinetic analyzer (SurPASS, Anton Paar, Austria) in a 0.001 mol L⁻¹ potassium chloride solution at pH from 4.1 to 9.7.

The specific areal resistance of the tested sub-4 nanometer porous membrane was measured in a 0.5 mol L⁻¹ NaCl solution via a four-compartment electric resistance analyzer (ChemJoy Polymer Material Co., Ltd., China). The pore size and molecular weight cutoff of the sub-4 nanometer porous membrane was determined by filtration of the solution containing 0.2 g L⁻¹ poly(ethylene glycol) polymers with different molecular weights. More details about the measurements of specific areal resistance and pore size of the tested membrane were given in the Supplementary Materials.

## Pressure-driven filtration performance

Pressure-driven filtration was conducted to measure its solute selectivity between dyes and inorganic salts using a cross-flow filtration cell. Firstly, a sub-4 nanometer porous membrane coupon as s tight ultrafiltration membrane (effective area: 22.9 cm²) was pre-compacted via

filtration of deionized water at 4 bar for 20 min. Afterward, filtration of pure inorganic salt (i.e., 0.5, 1.0, 5.0, 10.0, and 14.6 g L⁻¹ NaCl) or dye solutions (e.g., 0.1, 0.2, 0.4, 0.6, 0.8 and 1.0 g L⁻¹ reactive dye) was performed with the cross-flow rate of 66 L h⁻¹ at 25 ± 1 °C and 4 bar. Ultimately, filtration of reactive black 5/NaCl mixture solutions with different salt concentrations (i.e., up to 14.6 g L⁻¹) was performed at the same operation condition for determining the reactive black 5/NaCl selectivity of the sub-4 nanometer porous membrane. The rejection ($R_s$) of reactive black 5 or NaCl of the sub-4 nanometer porous membrane was determined by Eq. 1:

$$R_s = 1 - \frac{C_p}{C_0} \tag{1}$$

where $C_p$ and $C_0$ are the concentrations of reactive black 5 and NaCl in the permeate and feed, respectively.

The permeability ($J_p$) of the tested membrane during the pressure-driven filtration of different solutions was calculated by Eq. 2:

$$J_p = \frac{\Delta V}{A \cdot \Delta t \cdot \Delta P} \tag{2}$$

where $\Delta V$ is the volume the collected permeate at a fixed time interval ($\Delta t$); $A$ is the effective membrane filtration area; $\Delta P$ is the applied operation pressure.

The selectivity ($S_{NaCl/dye}$) between NaCl and reactive dye of the sub-4 nanometer porous membrane during the pressure-driven filtration of the dye/NaCl mixture solutions was determined by Eq. 3:

$$S_{NaCl/dye} = \frac{1 - R_{NaCl}}{1 - R_{dye}} \tag{3}$$

where $R_{NaCl}$ and $R_{dye}$ are the rejection of NaCl and reactive black 5 for the tested membrane, respectively, during the pressure-driven filtration of the reactive black 5/NaCl mixture solutions.

In order to separate reactive black 5 and NaCl from the dye/NaCl mixture solution for dye desalination and purification via pressure-driven filtration, a constant-volume diafiltration process using the sub-4 nanometer porous membrane as a tight ultrafiltration membrane was performed by the cross-flow filtration device at 4 bar (Supplementary Fig. 13)[38]. Specifically, a dye/NaCl mixture solution (i.e., 1.0 g L$^{-1}$ reactive black 5; 14.6 g L$^{-1}$ NaCl) was employed as the feed. Pure water with variable diavolumes (up to 4.6) was dripped into the feed to maintain the volume of the feed constant for NaCl removal from the dye/NaCl mixture solution.

The recovery efficiency ($\zeta$) of the dye in the constant-volume diafiltration process was determined by Eq. 4:

$$\zeta = 1 - \frac{C_{\text{loss, dye(p)}}}{C_{\text{feed, dye(p)}}} \quad (4)$$

where $C_{\text{loss, dye (p)}}$ is the final concentration of dye in the permeate during the constant-volume diafiltration process; $C_{\text{feed, dye (p)}}$ is the initial concentration of dye in the feed during the constant-volume diafiltration process.

The desalination efficiency ($\delta$) during the constant-volume diafiltration process was determined by Eq. 5:

$$\delta = 1 - \frac{C_{\text{final, salt(p)}}}{C_{\text{initial, salt(p)}}} \quad (5)$$

where $C_{\text{initial, salt (p)}}$ and $C_{\text{final, salt (p)}}$ are the initial and final concentrations of NaCl in the feed during the pressure-driven filtration process, respectively.

## Electro-driven filtration performance

Electro-driven filtration of the sub-4 nanometer porous membrane as anion conduction membrane for anion transfer, which was alternative to anion exchange membrane, was conducted in a custom-designed electrodialysis system (ChemJoy Polymer Material Co., Ltd., China) to separate the dye/NaCl mixture solution (Fig. 1A). This electro-driven filtration system was composed of a cathode, an anode, and an ion-exchange membrane stack (Supplementary Fig. 14A). Specifically, three cation exchange membranes and two sub-4 nanometer porous membranes (Supplementary Fig. 14B) were alternately placed in the membrane stack. The electrolyte solution containing 0.3 mol L$^{-1}$ Na$_2$SO$_4$ was used in both the cathode and anode chambers.

Firstly, 300 mL NaCl solutions with variable concentrations (i.e., 5.9, 8.8, 11.7, and 14.6 g L$^{-1}$) was employed as the feed at an applied direct current intensity of 0.5 A (with a recirculation flow rate of 10 L h$^{-1}$ at 25 ± 1 °C) to explore the anion transfer capability of the sub-4 nanometer porous membrane as anion conducting membrane for electro-driven desalination. The pure NaCl solution, which had the equivalent salinity with the feed, was employed in the concentrate chamber. Subsequently, different direct current intensities (i.e., 0.4, 0.5, 0.6, and 0.7 A) were applied to assess their effect on the desalination performance of the tested membrane in the electro-driven filtration system at the same operation condition. Once the conductivity of the feed dropped below the level of 0.8 mS cm$^{-1}$, the electro-driven filtration operation was terminated.

The desalination efficiency ($\varphi$) of the electro-driven filtration process was determined by Eq. 6:

$$\varphi = 1 - \frac{C_{\text{final, salt(e)}}}{C_{\text{initial, salt(e)}}} \quad (6)$$

where $C_{\text{initial, salt (e)}}$ and $C_{\text{final, salt (e)}}$ are the initial and final concentrations of NaCl in the feed during the electro-driven filtration process, respectively.

Finally, following the electro-driven filtration of pure NaCl solutions, the electrodialytic fractionation of the dye/NaCl mixture solution was conducted. Briefly, 300 mL dye/NaCl mixture solution (i.e., 1.0 g L$^{-1}$ reactive black 5; 14.6 g L$^{-1}$ NaCl) was employed as the feed in the electro-driven filtration process using the sub-4 nanometer porous as anion conducting membrane for fractionation of dye and NaCl at a current intensity of 0.6 A. During the fractionation of the dye/NaCl mixture solution, the electro-driven filtration process was stopped as the conductivity of the feed was lower than 1.4 mS cm$^{-1}$.

In addition, the antifouling property of the sub-4 nanometer porous membrane was explored with an eight-cycle continuous electrodialysis using the reactive black 5/NaCl mixture solution as the feed for one-step fractionation of the dye and NaCl at a current intensity of 0.6 A. On the other hand, a three-cycle continuous electrodialysis equipped with the commercial anion exchange membrane was performed under the same testing conditions for performance comparison.

The permselectivity ($P_{\text{dye}}^{\text{NaCl}}$) between NaCl and reactive dye of the sub-4 nanometer porous membrane during the electro-driven filtration process was determined by Eq. 7[48,49]:

$$P_{\text{dye}}^{\text{NaCl}} = \frac{\left(C_{0, \text{NaCl}} - C_{t, \text{NaCl}}\right) \cdot C_{t, \text{dye}}}{\left(C_{0, \text{dye}} - C_{t, \text{dye}}\right) \cdot C_{t, \text{NaCl}}} \quad (7)$$

where $C_{0,\text{NaCl}}$ and $C_{t,\text{NaCl}}$ are the initial and final concentration of NaCl in the feed, respectively; $C_{0,\text{dye}}$ and $C_{t,\text{dye}}$ are initial and final concentration of the reactive black 5 in the feed, respectively, during the electro-driven filtration process.

## Molecular dynamics simulation

To unravel the mass transfer mechanism of the sub-4 nanometer porous membrane in both the pressure-driven and electro-driven filtration processes, the transport behavior of different ions (i.e., Na$^+$, Cl$^-$ and dyes) was illustrated through molecular dynamics simulation. A more detailed description of the molecular dynamics simulation was given in the Supplementary Methods.

## Data availability

The authors declare that all data supporting the findings of this study are available within the paper and its supplementary information files or available from the corresponding author upon request.

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

## Acknowledgements

This work was financially supported by the National Natural Science Foundation of China (Grant No.: 22378389), the National Key Research

and Development Program of China (Grant No.: 2021YFC3201400), the National Natural Science Foundation of China-Regional Innovation Development Joint Fund (Grant No.: U24A2096), the Natural Science Foundation of Jiangxi Province (Grant No.: 20242BAB23019), the Natural Science Foundation of Fujian Province (2021J01628), a start-up fund for researchers of Jiangxi University of Science and Technology (Grant No.: 205200100721), the Royal Society (IEC\NSFC\211021, RG\R1\251471, IEC\NSFC\242089), the Royal Academy of Engineering (IF2223B–104) and the Leverhulme Trust (RPG-2022–177). D.H.S. acknowledges the support of the Korea Institute of Energy Technology Evaluation and Planning (KETEP) and the Ministry of Trade, Industry & Energy (MOTIE) of the Republic of Korea (No. 20224000000100). The molecular dynamics simulation for this article was (fully/partially) performed on resources of the National Supercomputing Center (NSCC), Singapore (https://www.nscc.sg).

## Author contributions

J.L. (J. Lin), K.Z., W.Y., M.X., and Y.W. conceived the idea and designed the research. J.L. (J. Lin), W.Y., Z.Y., T.C., J.H., L.C., and J.L. (J. Li) performed the pressure-driven and electro-driven filtration experiments. E.Y.M.A., W.T., W.P.C., T.Y.N., and D.H.S. performed the molecular dynamics simulation. J.L. (J. Lin), X.L., X.H., J.L. (J. Luo), D.H.S., S.Z., K.Z., M.X., W.Y., B.V.B., and Y.W. contributed to interpreting the data and writing the manuscript.

## Competing interests

The authors declare no competing interests.

## Additional information

[1]Jiangxi University of Science and Technology, Ganzhou, China. [2]Key Laboratory of Rare Earths, Ganjiang Innovation Academy, Chinese Academy of Sciences, Ganzhou, China. [3]School of Environment and Safety Engineering, Fuzhou University, Fuzhou, China. [4]State Key Laboratory of Biopharmaceutical Preparation and Delivery, Institute of Process Engineering, Chinese Academy of Sciences, Beijing, China. [5]Engineering Cluster, Singapore Institute of Technology, Singapore, Singapore. [6]School of Mechanical and Aerospace Engineering, Nanyang Technological University, Singapore, Singapore. [7]Institute of Energy Materials & Devices, Korea Institute of Energy Technology (KENTECH), Naju, Republic of Korea. [8]Institute for Frontier Materials, Deakin University, Geelong, Victoria, Australia. [9]HuiKang Advanced Institute of Technology, Shenyang, China. [10]Department of Chemical Engineering, University of Bath, Bath, United Kingdom. [11]Department of Chemical Engineering, Process Engineering for Sustainable Systems (ProcESS), KU Leuven, Leuven, Belgium. [12]Faculty of Engineering and the Built Environment, Department of Chemical, Metallurgical and Materials Engineering, Tshwane University of Technology, Pretoria, South Africa. ✉e-mail: kuo.zhong@hkait.cn; mx406@bath.ac.uk; ye0508@126.com; yhwan@gia.cas.cn

