## [Transparent Peer Review file · Nature Communications]

Sub-4 nanometer porous membrane enables highly efficient electrodialytic fractionation of dyes and inorganic salts

Corresponding Author: Professor Ming Xie

Version 0:

Reviewer comments:

Reviewer #1

(Remarks to the Author)

Lin et al devised a novel electro-driven filtration process using a sub-4 nanometer tight ultrafiltration membrane as alternative to conventional anion exchange membrane for dye desalination and purification. In which, they reported 98.15% desalination efficiency and 99.66% dye recovery for one-step fractionation of reactive dye and NaCl salt by the proposed process, markedly outperforming the system using commercial anion exchange membranes. It also exhibited high stability and efficiency of dye/salt without the trouble of membrane fouling. This work provides a new strategy to circumventing the membrane fouling in electro dialysis process for highly efficient dye/salt separation. Therefore, I'd like suggest its publication on Nat Commun. However, there are still some issues need to be revised before acceptance. The specific comments are as follows:

1. Some format issues should be addressed. Such as the expression of unit should be unified, g/L→g·L⁻¹.
2. How do the authors choose the sub-4 nanometer tight ultrafiltration membrane (TUF) as alternative to conventional anion exchange membrane? Does the pore size of TUF membrane affect this process? What are the main consideration factors?
3. FT-IR, XPS are suggested to identify the chemical composition of TUF membrane.
4. Figure 1e shows the TUF membrane is negatively charged at pH 7. How about the pH of the feed solution? Additionally, why the cations don't transmit across the TUF membrane from the concentrate cell to the dilute cell?
5. The numerical orders in figure 2 are not consistent with the description in main text.
6. The explanations and discussions of molecular dynamics simulation results are too perfunctory to make clear the relationship with experimental results. Authors even didn't mention the figures 2b, 5c and 5d.
7. Detailed operation conditions should be mentioned in main text or captions of figures 4a, 5a and other analogues.
8. Fig. 5a shows there is almost no membrane fouling occurred after 258 min operation. Extending operation time in each cycle may be more reasonable to confirm the antifouling performance of the process here in cyclic test (figure 6a). By the way, how to recycle the membrane?
9. The note (pure water) in figure S1a is incorrect.
10. There is no figures 5e and 5f (line 287).
11. High desalination ratio is achieved by the devised process. How about the treatment flux of this process? And a performance comparison on flux and separation factor is suggested with other processes. Such as loose NF.

Reviewer #2

(Remarks to the Author)

In this paper, a set of ultrafiltration + electro dialysis system was designed to treat the mixed solution of dye and salt, which effectively reduced the membrane fouling process under the premise of maintaining excellent desalination efficiency and dye recovery rate, and provided a new way for the treatment of salt-containing dye wastewater.

1. In the introduction part, the paper proposes that there will be a large amount of NaCl in the production and precipitation of dyes, and sodium sulfate will also exist in a large amount in the actual dye wastewater, and generally contain a variety of inorganic salts. Please verify and cite relevant literature.
2. Can the ultrafiltration and electro dialysis system used in this study effectively separate dyes and inorganic salts in the presence of multiple inorganic salts, and what are the effects of the types and concentrations of inorganic salts on the separation process? What is the effect of the presence of other anions (such as SO₄⁽²⁻⁾) on the separation?
3. Only one dye (RB5) was used in the study, but did the other dyes have a similar effect, how much of a role did the

aggregation effect of the dye play in the separation process, and what was the separation effect for the dyes with a less aggregation effect?

4. In Figure 6D, chloride ions diffuse from the anode to the cathode under the action of an electric field, why do the dye molecules with the same negative charge not accumulate on the surface of the film with the action of the electric field? Because whether it is a simple ultrafiltration or ultrafiltration + electro dialysis system, there is charge repulsion on the negatively charged dye and negatively charged membrane surface, and the pressure of the ultrafiltration process alone will pollute the membrane, why will the electro dialysis process not pollute the membrane under the action of the electric field force (the negatively charged dye is forced to the membrane surface and migrates to the membrane surface)?

5. Although the ultrafiltration process alone will be subject to more serious membrane fouling, the whole filtration system is relatively simple, and the membrane fouling can reduce the harm of membrane fouling through regular membrane cleaning. However, the ultrafiltration + electro dialysis system is more complex (requires an electric field, requires an additional cation exchange membrane), and in terms of overall cost and economic benefits, is there a better overall effect in this study?

6. Figures 5E and 5F mentioned in the discussion do not exist in the figures.

7. Figure S10 B is incorrect and inconsistent with the description "commercial anion exchange membrane declined from 92.87% to 84.98% after the three-cycle electro dialytic filtration operation".

Version 1:

Reviewer comments:

Reviewer #1

(Remarks to the Author)

The authors have satisfactorily addressed the reviewers' concerns and conducted sufficient supplementary research and revisions to the manuscript. Therefore, I recommend acceptance for publication.

Reviewer #2

(Remarks to the Author)

The paper is well revised and concerns are replied properly.

Response to the Reviewers

The authors appreciate the Editor and Reviewers for their constructive comments and suggestions on our revised manuscript. The followings are our point-by-point responses to Reviewers' comments. All corrections in the manuscript are marked in **RED FONT**.

Reviewer #1 (Remarks to the Author):

Lin et al devised a novel electro-driven filtration process using a sub-4 nanometer tight ultrafiltration membrane as alternative to conventional anion exchange membrane for dye desalination and purification. In which, they reported 98.15% desalination efficiency and 99.66% dye recovery for one-step fractionation of reactive dye and NaCl salt by the proposed process, markedly outperforming the system using commercial anion exchange membranes. It also exhibited high stability and efficiency of dye/salt without the trouble of membrane fouling. This work provides a new strategy to circumventing the membrane fouling in electrodialysis process for highly efficient dye/salt separation. Therefore, I'd like suggest its publication on Nat Commun. However, there are still some issues need to be revised before acceptance. The specific comments are as follows:

Response to Reviewer comment: We thank the reviewer for his/her positive assessment and constructive comments. We will follow the suggestions from the reviewer to carefully revise the manuscript.

1. Some format issues should be addressed. Such as the expression of unit should be unified, $\text{g/L} \rightarrow \text{g} \cdot \text{L}^{-1}$.

Response to Reviewer comment No. 1: We appreciate the reviewer for this comment. In the revised manuscript, we follow the suggestions from the reviewer to unify all the unit of "g/L" into " $\text{g} \cdot \text{L}^{-1}$ ". Furthermore, other units (such as $\text{LMH} \cdot \text{bar}^{-1}$, mS/cm , g/L/h , A/fs and mg/L) are revised at a unified mode ($\text{L} \cdot \text{m}^{-2} \cdot \text{h}^{-1} \cdot \text{bar}^{-1}$, $\text{mS} \cdot \text{cm}^{-1}$, $\text{g} \cdot \text{L}^{-1} \cdot \text{h}^{-1}$, $\text{A} \cdot \text{fs}^{-1}$ and $\text{mg} \cdot \text{L}^{-1}$) carefully through the whole manuscript and Supplementary information file. The reviewer can see two examples below (marked in red):

**Fig. 6** Eight-cycle electrodialytic filtration using the sub-4 nanometer porous
 membrane as anion conducting membrane for fractionation of dye and NaCl from
 the reactive black 5/NaCl mixture solution. (A) Evolution of conductivity in the
 concentrate and diluate, (B) NaCl content in the diluate and desalination efficiency, (C)
 Dye content in the concentrate and recovery rate, (D) Illustration of anion (Cl^- and
 reactive black 5) transfer of the sub-4 nanometer porous membrane for electrodialytic
 fractionation of the reactive black 5/NaCl mixture solution.

**Fig. S7 Electrodialytic filtration performance of the sub-4 nanometer porous**
 **membrane as a role of anion conducting membrane in pure NaCl solutions with**
 **different salinities. (A) Evolution of conductivity in both concentrate and diluate**
 **solutions, (B) NaCl transfer rate, (C) NaCl concentration in the diluate and desalination**
 **efficiency.**

2. How do the authors choose the sub-4 nanometer tight ultrafiltration membrane (TUF)
 as alternative to conventional anion exchange membrane? Does the pore size of TUF
 membrane affect this process? What are the main consideration factors?

**Response to Reviewer comment No. 2:** We thank the reviewer for this insightful
 comment. In this study, we focus on the fractionation of dye and salt, which requires a
 membrane that exhibits high dye rejection while allowing salt to pass through. Notably,
 during the pressure-driven filtration process, we found that the sub-4 nanometer TUF
 membrane has an impressive dye rejection (due to the dye cluster caused by aggregation
 via hydrophobic interaction) and low salt rejection (see Figure 2 below), showing a
 considerably high selectivity between dye and salt (see updated Supplementary Fig. S7

below). Therefore, it can be an ideal candidate for anion conducting membrane as
 alternative to conventional anion exchange membrane. The membrane's nanochannels
 efficiently retain dye molecules through a size exclusion mechanism, while
 concurrently providing ample pathways for Cl^- ion transport under an applied electric
 field. This dual functionality enables a one-step, electro-driven fractionation process
 (see Figure 6D), underscoring the membrane's potential in practical applications.

 **Fig. 2 Filtration performance of the sub-4 nanometer porous membrane in**
 **reactive black 5/NaCl mixture solution with different salinities. (A) Rejection of**
 **solute (reactive black 5 and NaCl) and permeability.**

 **Fig. S7. The selectivity between NaCl and reactive black 5 of the sub-4 nanometer**
 **porous membrane during the pressure-driven filtration of the dye/NaCl mixture**
 **solutions at different salinities**

**Fig. 6 (D)** Illustration of anion (Cl^- and reactive black 5) transfer of the sub-4 nanometer
 porous membrane for electrodralytic fractionation of the reactive black 5/ NaCl mixture
 solution.

The pore size of the TUF membranes is indeed critical for the effective rejection of dye
 molecules during electro-driven separation of dye/salt mixtures. Actually, we
 previously tested the dye separation performance of four polyethersulfone membranes
 with different pore sizes in the reactive black 5 solution, including nanofiltration
 membrane M1 (molecular weight cutoff of ~ 500 Da), Sub-4 nm TUF membrane M2
 (molecular weight cutoff of 2292 Da) in this work, TUF membrane M3 (molecular
 weight cutoff of ~ 4000 Da) and TUF membrane M4 (molecular weight cutoff of ~ 5000
 83 Da). The dye rejection performance of these four tested polyethersulfone membranes
 (M1, M2, M3 and M4) is shown in the **Figure A1** below:

Figure A1. Rejection of reactive black 5 using the commercial polyethersulfone membranes with different pore sizes (Nanofiltration membrane M1: molecular weight cutoff of ~500 Da; Sub-4 nm TUF membrane M2: molecular weight cutoff of 2292 Da; TUF membrane M3: molecular weight cutoff of ~4000 Da; TUF membrane M4: molecular weight cutoff of ~5000 Da) at a pressure of 4 bar.

As indicated in the **Figure A1**, we can conclude that the membranes with larger pore sizes exhibit lower dye rejection, due to their diminished size exclusion effect for reduced dye rejection. Specifically, the nanofiltration membrane M1 (MWCO of ~500 Da) shows a rejection of 99.94% for reactive black 5, while the sub-4 nm TUF membrane (M2, MWCO of 2292 Da) has a comparably high dye rejection (99.93%) with the nanofiltration membrane M1. However, the M3 (MWCO of ~4000 Da) and M4 membrane (MWCO of ~5000 Da) with larger pore sizes have a lower dye rejection (99.31% and 98.66% for rejection of reactive black 5, respectively), which can significantly compromise dye recovery during the electro-driven separation process when they are used as anion conducting membrane for one-step dye/salt separation. Such results demonstrate that the sub-4 nm tight ultrafiltration membrane used in the study is sufficient for dye retention and recovery during electro-driven separation. Thus, selecting a membrane with an appropriately small pore size is essential, which is why

we chose the sub-4 nanometer TUF membrane (molecular weight cutoff of 2,292 Da)
for our study.

Furthermore, the reviewer's comments have inspired us to consider further
enhancements to the electro-driven filtration process and the development of advanced
anion-conducting membranes. Currently, our tight ultrafiltration membranes are
polyethersulfone-based and fabricated via non-solvent phase inversion, resulting in a
spongy structure with a relatively thick selective layer (~500 nm thickness; see Figure
1C). To improve the electro-driven performance, we plan to develop thin-film tight
ultrafiltration membranes using interfacial polymerization. This approach will allow us
to reduce the selective layer thickness to ~100 nm, thereby shortening the anion transfer
pathway and enhancing the anion transfer rate. Additionally, by varying the oil/water
phase monomers during the fabrication of the thin-film tight ultrafiltration membrane,
we will tailor the membrane surface charge and hydrophilicity to further facilitate anion
transport by the electro-driven separation mode.

In future work, we will fabricate these next-generation membranes with a focus on
optimizing the pore size, selective layer thickness, surface charge, and hydrophilicity
to significantly improve the electro-driven separation performance for dye/salt
mixtures. We are excited about the potential of these developments and sincerely
appreciate the reviewer's valuable input that has guided us in this promising direction.

**Fig. 1 (C)** Cross-sectional SEM images of the sub-4 nanometer porous membrane.

3. FT-IR, XPS are suggested to identify the chemical composition of TUF membrane.

**Response to Reviewer comment No. 3:** We thank the reviewer a lot for this comment.
 Now we added the FTIR and XPS measurement for TUF membrane with the discussion.
 The reviewer can see the FTIR and XPS measurement with the detailed discussion in
 manuscript and supplementary information file (Updated Supplementary Fig. S1)
 below:

2. Supplementary data and figures

2.1 Chemical composition of the membrane

 **Fig. S1 Chemical composition measurement of the tested membrane. (A) FTIR;**
 **(B) XPS wide scan and high-resolution scan spectra for C 1s, O 1s and S 2p.**

The characteristic stretching vibrations of polyethersulfone (i.e., at the wavenumber of
1576.5, 1484.4, 1411.1, 1321.0, 1296.6, 1236.6, 1147.4, 1103.6 and 717.4 cm^{-1}) can be
observed in the FTIR spectra of the tested membrane in this study. Specifically, three
typical peaks at the wavenumbers of 1576.5, 1484.4 and 1411.1 cm^{-1} are assigned to
aromatic ring groups (benzene rings)¹¹; the intrinsic peaks at the wavenumbers of
1321.0/1296.6 cm^{-1} and 1147.4/1103.6 cm^{-1} are attributed to the asymmetric and
symmetric stretching vibrations of sulfone groups (Ar-SO₂-Ar group), respectively¹¹;
the wavenumber of 717.4 cm^{-1} represents the stretching vibrations of C-S-C group¹²; a
strong peak at the wavenumber of 1236.6 cm^{-1} is associated to aromatic ether structure
(Ar-O-Ar group)^{11,12}; in addition, the wavenumbers of 3095.7, 3068.2, 835.5 and 700.5
149 cm^{-1} are due to the stretching vibration of C-H group¹³.

**Results (In the main text of the manuscript)**

**Characterization of nanoporous membrane**

The morphology of the membrane is critical for solute rejection in both pressure-driven
and electro-driven filtration. The tested membrane had no visible pores on the surface,
as observed by SEM (**Fig. 1B**). In particular, the sub-4 nm pore size of the tested
membrane represented a tight nanoscale surface structure. Furthermore, the tested
membrane showed a skin layer with a thickness of ~500 nm (**Fig. 1C**), which provided
an MWCO of 2,292 Da with mean effective pore diameter of 1.52 nm (**Fig. 1D**), as
measured from the rejection of different molecular-weight polyethylene glycol
polymers. The cross-section SEM image illustrated that the sub-4 nanometer porous
membrane established a hierarchical porous spongy structure, which would offer a
strong mechanical support during both pressure-driven and electro-driven filtration
process. Furthermore, the FTIR measurement of the tested membrane shows the
intrinsic peaks at the wavenumbers of 1576.5/1484.4/1411.1,
1321.0/1296.6/1147.4/1103.6, 1236.6 and 717.4 cm^{-1} , which are associated with the
stretching vibration of aromatic ring groups (benzene rings), sulfone groups, aromatic
ether structure, and C-S-C group, respectively (**Supplementary Fig. S1A**)^{40, 41}. The
characteristic stretching vibrations demonstrated that the tested membrane is the
polyethersulfone-based membrane, which can be further confirmed by the XPS scan
spectra of C 1s, O 1s and S 2p (**Supplementary Fig. S1B**)^{42, 43}.

4. Figure 1e shows the TUF membrane is negatively charged at pH 7. How about the
pH of the feed solution? Additionally, why the cations don't transmit across the TUF
membrane from the concentrate cell to the dilute cell?

**Response to Reviewer comment No. 4:** Our zeta potential analysis (Figure 1E, see
below) conclusively establishes that the TUF membrane retains a robust negative
surface charge across a wide pH spectrum (pH range of 4.1-9.7), including the neutral
pH 7. Critically, the feed solution containing reactive black 5 and NaCl was measured
at pH 5.9, a value well within the range where the membrane's negative charge remains
stable. This confirms that the sub-4 nanometer TUF membrane maintains its
electrostatic repulsion characteristics under the experimental conditions, directly
influencing its interaction with anionic species in the feed (e.g., reactive black 5 dye
and Cl⁻ ions).

Figure 1E. Zeta potential of the tested TUF membrane

**Figure 1A** below in the manuscript clearly illustrates the electro-driven ion transfer
mechanism within our membrane stack, which is configured as “Anode / cation
exchange membrane (CEM) / anion-conducting TUF membrane / CEM / Cathode.”
Under an applied electric field, Cl⁻ ions migrate from the diluate chamber to the
concentrate chamber exclusively through the nanochannels of the TUF membrane. This
selective anion transport is enabled by the TUF membrane's unique structure and is
further supported by the cation exchange membranes (CEMs), which effectively block
Cl⁻ ions in the electrolyte chamber due to their inherent charge-repulsion properties.

Meanwhile, Na^+ ions in the electrolyte chamber transfer into the concentrate chamber
 through the CEM, and Na^+ ions from the diluate chamber concurrently move into the
 electrolyte chamber to maintain a consistent sodium concentration. This coordinated
 ion movement ensures that, to uphold charge neutrality in the concentrate chamber, Na^+
 ions do not back-transfer through the TUF membrane into the diluate chamber.

As a result, the desalination of the diluate (a reactive dye/ NaCl mixture) is achieved
 without membrane fouling, markedly outperforming conventional electro-dialysis
 processes. This mechanism underscores the intrinsic advantage of our electro-driven
 filtration system, where the TUF membrane functions as an efficient anion-conducting
 membrane.

**Fig. 1 (A)** stack configuration of the novel electro-driven filtration process

5. The numerical orders in figure 2 are not consistent with the description in main text.

**Response to Reviewer comment No. 5:** We make a deep apology to the reviewer for
 this mistake. We thank the reviewer very much to point out the mistake. Now we correct
 the numerical orders in Figure 2 to make it consistent with the description in the main

text for avoiding the misunderstanding to the reviewer and potential readers. The
reviewer can see the updated number orders in Figure 2 below:

Generally, a dye-containing waste liquor with high salinity can be generated during the
synthesis of reactive dyes. The co-existence of NaCl with dyes could potentially affect
the filtration behavior of the sub-4 nm membrane (**Fig. 2**). For example, when the NaCl
concentration in the reactive black 5/NaCl mixture solution increased from 0 to 14.6
$\text{g}\cdot\text{L}^{-1}$, the corresponding rejection of reactive black 5 decreased (**Fig. 2A**), likely due to
the intensification in electrostatic shielding effect caused by the elevated salt
concentration, diminishing the electrostatic repulsion against the dyes. Simultaneously,
the membrane swelling caused by the elevated salt concentration, inducing a slight
increase in pore size of membrane surface, could potentially facilitate the transfer of the
dye through the sub-4 nanometer porous membrane. On the other hand, the formation
of dye clusters via hydrophobic interaction of dye molecules can partially offset the
negative effect of elevated NaCl concentration on the dye rejection, yielding a slight
decline in dye rejection from $99.93\pm 0.02\%$ to $98.09\pm 0.22\%$ with increasing salt
concentration (**Fig. 2A**). This can be demonstrated by the consistently high dye
rejection in the molecular dynamics simulation during the filtration of dye/NaCl
mixture solution (**Fig. 2B**, **Fig. 2C** and **Supplementary Movie S3**). However, the
selectivity between NaCl and reactive black 5 was declined drastically from 990.2 to
52.0 with increasing NaCl concentration (**Supplementary Fig. S7**), mainly due to the
fact that the electrostatic shielding effect on the membrane surface was amplified with
increasing NaCl concentration, which undermines the electrostatic repulsion with
reduced dye retention, resulting in dye loss (i.e., 1.71%) through the membrane
permeate and thus compromising the dye/salt fractionation during the pressure-driven
constant-volume diafiltration process for dye purification (**Fig. 3** and **Supplementary**
**Table S1**). On the other hand, the permeability of the membrane also showed a
significant decrease with increasing NaCl concentration (up to $14.6 \text{ g}\cdot\text{L}^{-1}$) by 44.0%,
declining from $17.73\pm 0.26 \text{ L}\cdot\text{m}^{-2}\cdot\text{L}^{-1}\cdot\text{bar}^{-1}$ to $9.92\pm 0.19 \text{ L}\cdot\text{m}^{-2}\cdot\text{L}^{-1}\cdot\text{bar}^{-1}$ (**Fig. 2A**),
which can markedly deteriorate the dye/salt fractionation efficacy through pressure-
driven filtration.

6. The explanations and discussions of molecular dynamics simulation results are too
perfunctory to make clear the relationship with experimental results. Authors even
didn't mention the figures 2b, 5c and 5d.

**Response to Reviewer comment No. 6:** We apologize for the numbering error in the
discussion section, where Figures 5c and 5d were mistakenly labelled as Figures 5E and
5F, respectively.

To clarify, in this manuscript, Figure 2b and Figure 5C describe the details of the
molecular dynamic simulation domain for pressure-driven and electro-driven filtration
process, respectively. Although the detailed simulation methodology is provided in the
Supporting Information, we have carefully discussed the simulation outcomes in
Figures 2C and 5D, which demonstrate excellent agreement with our experimental
results.

To address the reviewer's concerns and enhance clarity, we have updated and
strengthened the discussion regarding Figures 2B, 2C, 5C, and 5D, as well as
Supplementary Movie S5. In this study, we used the molecular dynamics simulation in
Figure 2C and Figure 5D to demonstrate the good consistency of the experiment results
with the molecular dynamics simulation results. Now we update and strengthen the
discussion on Figure 2b, Figure 2C, Figure 5c and Figure 5D and Supplementary Movie
S5 of the molecular dynamics simulation in the manuscript to ensure that the reviewers
and the potential readers can see the simulation process and simulation results clearly
(in the Section of "Pressure-driven filtration performance" and "Discussion", marked
in red in the manuscript). The reviewer can see the revision detailed below:

**Pressure-driven filtration performance**

Generally, a dye-containing waste liquor with high salinity can be generated during the
synthesis of reactive dyes. The co-existence of NaCl with dyes could potentially affect
the filtration behavior of the sub-4 nm membrane (**Fig. 2**). For example, when the NaCl
concentration in the reactive black 5/NaCl mixture solution increased from 0 to 14.6
g·L⁻¹, the corresponding rejection of reactive black 5 decreased (**Fig. 2A**), likely due to
the intensification in electrostatic shielding effect caused by the elevated salt
concentration, diminishing the electrostatic repulsion against the dyes. Simultaneously,
the membrane swelling caused by the elevated salt concentration, inducing a slight

increase in pore size of membrane surface, could potentially facilitate the transfer of the
dye through the sub-4 nanometer porous membrane. On the other hand, the formation
of dye clusters via hydrophobic interaction of dye molecules can partially offset the
negative effect of elevated NaCl concentration on the dye rejection, yielding a slight
decline in dye rejection from $99.93\pm 0.02\%$ to $98.09\pm 0.22\%$ with increasing salt
concentration (Fig. 2A). This can be demonstrated by the consistently high dye
rejection in the molecular dynamics simulation during the filtration of dye/NaCl
mixture solution (Fig. 2B, Fig. 2C and Supplementary Movie S3)

Discussion

The tested sub-4 nanometer membrane established an enhanced anion electrodynamic
transport via its intrinsically nanoporous structure. Molecular dynamic simulations of
anion transport were further conducted to illustrate the ion transfer mechanism of the
sub-4 nanometer porous membrane as anion conducting membrane under a direct
current field (Fig. 5C, Fig. 5D and Supplementary Movies S5). Due to the direct
current field implementation across the cation exchange membrane and sub-4
nanometer porous membrane, Cl^- ions in the feed were forced to migrate through the
nanochannels of the sub-4 nanometer porous membrane at a drift velocity of $2.7\text{E-}4$
$\text{\AA}\cdot\text{fs}^{-1}$ (Fig. 5D); on the other hand, Na^+ ions in the feed were forced to migrate through
the cation exchange membrane at the opposite direction for electrodynamic desalination
of the reactive black 5/NaCl mixture solution. In the contrary, the movement of the
reactive black 5 is less affected by the applied direct current field. Specifically, reactive
black 5 molecules, with larger size, were observed to have an extremely low drift
velocity (i.e., $1.1\text{E-}5 \text{\AA}\cdot\text{fs}^{-1}$) for moving towards the sub-4 nanometer porous membrane
(Fig. 5D), which is much less than Cl^- ions. Therefore, based on the large difference in
drift velocity between reactive black 5 and NaCl, the directed transfer status for both
reactive black 5 and Cl^- allows for their selective fractionation. Moreover, with the aid
of the enhanced size exclusion effect, reactive black 5 was sufficiently intercepted by
the sub-4 nanometer porous membrane for effective one-step fractionation of the
reactive dye and Cl^- ions (Supplementary Movies S5).

7. Detailed operation conditions should be mentioned in main text or captions of figures
4a, 5a and other analogues.

**Response to Reviewer comment No. 7:** We greatly appreciate the reviewer for further
improve the readability of the figure captions. As advised by the reviewer, we have
added the detailed operation condition of the membrane during the pressure-driven
filtration and electro-driven filtration process at the section of “Methods”. The reviewer
can see the detailed operation conditions at the section of “Methods” (subsections of
“**Pressure-driven filtration performance**” and “**Electro-driven filtration**
**performance**”), which are shown below:

**Pressure-driven filtration performance**

Pressure-driven filtration was conducted to measure its solute selectivity between dyes
and inorganic salts using a cross-flow filtration cell. Firstly, a sub-4 nanometer porous
membrane coupon as a tight ultrafiltration membrane (effective area: 22.9 cm²) was
pre-compacted via filtration of deionized water at 4 bar for 20 min. Afterwards,
filtration of pure inorganic salt (i.e., 0.5, 1.0, 5.0, 10.0 and 14.6 g·L⁻¹ NaCl) or dye
solutions (i.e., 0.1, 0.2, 0.4, 0.6, 0.8 and 1.0 g·L⁻¹ reactive black 5) was performed **with**
**the cross flow rate of 66 L·h⁻¹ at 25±1°C and 4 bar**. Ultimately, filtration of reactive
black 5/NaCl mixture solutions with different salt concentrations (i.e., up to 14.6 g·L⁻¹
¹) was **performed at the same operation condition** for determining the reactive black
5/NaCl selectivity of the sub-4 nanometer porous membrane.

**Electro-driven filtration performance**

Firstly, 300 mL NaCl solutions with variable concentrations (i.e., 5.9, 8.8, 11.7 and 14.6
g·L⁻¹) was employed as the feed at an applied direct current intensity of 0.5 A (**with**
**recirculation flow rate of 10 L·h⁻¹ at 25±1°C**) to explore the anion transfer capability of
the sub-4 nanometer porous membrane as anion conducting membrane for electro-
driven desalination. The pure NaCl solution, which had the equivalent salinity with the
feed, was employed in the concentrate chamber. Subsequently, different direct current
intensities (i.e., 0.4, 0.5, 0.6 and 0.7 A) were applied to assess their effect on the
desalination performance of the tested membrane in the electro-driven filtration system
**at the same operation condition**. Once the conductivity of the feed dropped below the
level of 0.8 mS·cm⁻¹, the electro-driven filtration operation was terminated.

Additionally, the antifouling property of the sub-4 nanometer porous membrane was
explored with an eight-cycle continuous electro-dialysis using the reactive black 5/NaCl
mixture solution as the feed for one-step fractionation of the dye and NaCl at a current

intensity of 0.6 A. On the other hand, a three-cycle continuous electro dialysis equipped
with the commercial anion exchange membrane was performed under **the same testing**
**condition** for performance comparison.

8. Fig. 5a shows there is almost no membrane fouling occurred after 258 min operation.
Extending operation time in each cycle may be more reasonable to confirm the
antifouling performance of the process here in cyclic test (figure 6a). By the way, how
to recycle the membrane?

**Response to Reviewer comment No. 8:** We thank the reviewer very much for this
comment. Fig. 5A and Fig. 5B show the electro-driven separation performance of the
tight ultrafiltration membrane in dye/salt mixture solution, demonstrating an excellent
efficiency for dye/salt fractionation (98.15% desalination efficiency, 99.66% dye
recovery in Supplementary Table S1, see Table S1 below). We found that the tight
ultrafiltration membrane as anion conducting membrane had the strong repulsion effect
against the dye, since the tight ultrafiltration membrane can effectively repel the dye
from entering the membrane pore structure based on the size exclusion and electrostatic
repulsion effect (the membrane is negatively charged, which can effectively repel the
reactive dye with negative charge through electrostatic repulsion effect). Therefore, we
extending operation time in each cycle to further evaluate the antifouling performance
of the membrane in cyclic test. From Figure 6A, we can see the identical decay in feed
conductivity and increase in the concentrate conductivity in each cycle operation during
this eight-cycle electro dialytic filtration, demonstrating the consistent fractionation
performance of the tight ultrafiltration membrane as anion conducting membrane for
reactive black 5 and NaCl. This continuous eight-cycle electro dialytic filtration featured
a preponderant long-term stability of the sub-4 nanometer porous membrane for
steadily exceptional desalination (desalination efficiency of 98.09%-98.22%) and dye
recovery (99.63%-99.69%) (Figure 6B and Figure 6C, see below as well),
demonstrating an impressive antifouling property, which can be reflected by the slight
boost in specific areal membrane resistance after the eight-cycle electro dialytic
filtration (updated Supplementary Fig. S11 below).

Actually, since the impressive antifouling performance of the TUF membrane as anion
conducting membrane, we recycled the membrane using the simple water flushing after

one-cycle electro-driven filtration process to remove the dye species in the membrane
 stack.

 **Fig. 5** Electro-dialytic filtration performance of the sub-4 nanometer porous
 membrane as a role of anion conducting membrane in the reactive black 5/NaCl
 mixture solution for fractionation of dye and NaCl. (A) Evolution of conductivity in
 both the concentrate and diluate, (B) NaCl content in the diluate and dye content in the
 concentrate.

**Table S1.** Performance of the pressure-driven diafiltration and electro-driven filtration
 process for fractionation of reactive dye and NaCl using the sub-4 nanometer porous
 membrane

Mode	NaCl content in feed, g.L ⁻¹	Desalination efficiency, %	Recovery of reactive dye, %
Pressure-driven diafiltration	0.28	98.07	98.29
Electro-driven filtration	0.27	98.15	99.66

**Fig. 6** Eight-cycle electrodialectic filtration using the sub-4 nanometer porous
 **membrane as anion conducting membrane for fractionation of dye and NaCl from**
 **the reactive black 5/NaCl mixture solution.** (A) Evolution of conductivity in the
 **concentrate and diluate,** (B) **NaCl content in the diluate and desalination efficiency,** (C)
 **Dye content in the concentrate and recovery rate.**

**Fig. S11. Specific areal electric resistance of sub-4 nanometer porous membrane**
**before and after fouling during an eight-cycle electro dialytic filtration of the**
**reactive dye/NaCl mixture solution.**

9. The note (pure water) in figure S1a is incorrect.

**Response to Reviewer comment No. 9:** We thank the reviewer very much. We also
apologize to the reviewer that we made a mistake when we edited the previous figure
S1. The note “Pure water” should be placed in the permeability part of Figure S2B.
Now we follow the suggestion from the reviewer and update the Figure S2B. The
reviewer can see that below as well:

**Fig. S2. Pressure-driven filtration performance of the sub-4 nanometer porous**
**membrane in pure NaCl solution with different salinities. (B) Permeability.**

10. There is no figures 5e and 5f (line 287).

**Response to Reviewer comment No. 10:** We are sorry that we made a typo for that.
Actually, they are figures 5c and 5d, rather than figure 5e and 5f. Now we revise
“figures 5e and 5f” into “figures 5C and 5D” in the section of “Discussion” in the
manuscript to avoid the misunderstanding to the reviewer and the potential readers. The
reviewer can see that below as well:

**Discussion**

The tested sub-4 nanometer membrane established an enhanced anion electro-dialytic
transport via its intrinsically nanoporous structure. Molecular dynamic simulations of
anion transport were further conducted to illustrate the ion transfer mechanism of the
sub-4 nanometer porous membrane as anion conducting membrane under a direct
current field (**Fig. 5C, Fig. 5D** and **Supplementary Movies S5**).

11. High desalination ratio is achieved by the devised process. How about the treatment
flux of this process? And a performance comparison on flux and separation factor is
suggested with other processes. Such as loose NF.

**Response to Reviewer comment No. 11:** We thank the reviewer very much for this
comment. Indeed, the desalination ratio is high for the devised process. Our study
introduces a novel concept by demonstrating that tight ultrafiltration membranes can
function effectively as anion conducting membranes for the electro-driven fractionation
of dye and salt. This approach not only validates the practicability of using these
membranes in an electro-driven mode but also broadens their potential applications
beyond traditional pressure-driven processes. We believe our work will inspire further
exploration and innovation in the field of membrane separation technologies. Fig. 6 in
the manuscript demonstrates the potential advantages of the devised electro-driven
filtration process using tight ultrafiltration membrane as anion conducting membrane
for effective fractionation of dye and salt without membrane fouling, which markedly
outperforms the conventional electro-dialysis (conventional electro-dialysis suffers from
the severe fouling of commercial anion exchange membrane caused by dye adsorption
in updated Supplementary Fig. S12, see below) and pressure-driven ultrafiltration
membrane process (pressure-driven ultrafiltration requires constant-volume
diafiltration procedure with a large consumption of pure water for fractionation of dye
and salt. And the salt in the permeate can be diluted to a very low concentration level,
which fails to be recycled for reuse).

**Fig. S12. Three-cycle electrodialytic filtration using commercial anion exchange**
 **membrane for fractionation of dye and NaCl from the reactive black 5/NaCl**
 **mixture solution. (A)** Evolution of conductivity in the concentrate and diluate
 solutions, **(B)** NaCl content in the diluate and desalination efficiency, **(C)** Illustration
 of anion (Cl^- and reactive black 5) transfer of commercial anion exchange membrane
 during electrodialytic fractionation of the reactive black 5/NaCl mixture solution, **(D)**
 Specific areal electric resistance of commercial anion exchange membrane before and
 after fouling.

Currently, it is not easy to evaluate the treatment flux of the electro-driven filtration
 process using this sub-4 nm tight ultrafiltration membrane as anion conducting
 membrane. Since we separate the dye and salt in the electro-driven process, we concern
 about the electro-driven anion transfer rate for desalination, and we don't need to care
 about the flux of the membrane at the pressure-driven mode. In future, we need to
 optimize the stack configuration and operation conditions systematically to precisely
 evaluate the treatment efficiency of the electro-driven filtration process in practical
 applications.

In this work, we have conducted a comparison of anion transfer rate between sub-4 nm
 TUF membrane (molecular weight cutoff of 2292 Da) and a commercial loose
 nanofiltration membrane (molecular weight cutoff of ~500 Da) as anion conducting
 membranes in the pure NaCl solution ($\sim 14.6 \text{ g}\cdot\text{L}^{-1}$) during the electro-driven separation
 process (see the **Figure A2** below). As shown in **Figure A2** below, the loose
 nanofiltration membrane with ~ 500 Da MWCO requires a longer operation time to
 achieve the similar desalination efficiency (**Figure A2A** and **Figure A2B**), compared
 to the sub-4 nm TUF membrane during electro-driven separation process, which can be
 reflected by the slower decay in conductivity of the feed (diluate). Since the loose
 nanofiltration membrane with ~ 500 Da MWCO has a small pore size, which provides
 insufficient nanochannels for reduced anion transfer. Specifically, the loose
 nanofiltration membrane with ~ 500 Da MWCO has a NaCl transfer rate of $3.23 \text{ g}\cdot\text{L}^{-1}\cdot\text{h}^{-1}$
 $\text{L}^{-1}\cdot\text{h}^{-1}$, which is much lower than the sub-4 nm TUF membrane ($3.99 \text{ g}\cdot\text{L}^{-1}\cdot\text{h}^{-1}$) at the
 current intensity of 0.5 A (**Figure A2C**). Therefore, the sub-4 nm TUF membrane has
 the better electro-driven anion transfer capacity, which remarkably outperforms the
 loose nanofiltration membrane.

**Figure A2. Electrodialytic filtration performance of the sub-4 nanometer porous**
 **membrane (MWCO of 2292 Da) and loose nanofiltration membrane (MWCO of**

**~500 Da) as a role of anion conducting membrane in pure NaCl solution (~14.6**
**g·L⁻¹) at current intensity of 0.5 A. (A) Evolution of conductivity in both**
**concentrate and diluate solutions, (B) NaCl concentration in the diluate and**
**desalination efficiency, (C) NaCl transfer rate.**

Currently, we used polyethersulfone-based tight ultrafiltration membrane as anion
conducting membrane. However, the polyethersulfone-based tight ultrafiltration
membrane in this study has the spongy structure with a thick selective layer (thickness
of ~500 nm). Therefore, the anion transfer will be partially hindered. Therefore, in
future, we will design the new thin-film tight ultrafiltration membranes with ultra-thin
selective layers (thickness of ~100 nm) through interfacial polymerization method,
which can remarkably shorten the ion transfer pathway and reduce the resistance against
anion for facilitating the anion transfer and thus enhancing the treatment efficiency.
Finally, we will optimize the stack configurations and operation conditions to assess
the treatment flux of the electro-driven filtration process. We thank the reviewer again!

**Reviewer #2 (Remarks to the Author):**

In this paper, a set of ultrafiltration + electro dialysis system was designed to treat the
mixed solution of dye and salt, which effectively reduced the membrane fouling process
under the premise of maintaining excellent desalination efficiency and dye recovery
rate, and provided a new way for the treatment of salt-containing dye wastewater.

**Response to Reviewer comment:** We thank the reviewer for his/her positive
assessment of our manuscript and constructive comments. We will follow the
suggestions from the reviewer to carefully revise the manuscript.

1. In the introduction part, the paper proposes that there will be a large amount of NaCl
in the production and precipitation of dyes, and sodium sulfate will also exist in a large
amount in the actual dye wastewater, and generally contain a variety of inorganic salts.
Please verify and cite relevant literature.

**Response to Reviewer comment No. 1:** We greatly thank the reviewer for this
comment. As suggested by the reviewer, we have carefully verified the background of
sodium sulfate and added relevant literature which is related to both Na₂SO₄ and NaCl
(with the references). The reviewer can see that below:

During the dye synthesis process, acid and alkali would be subsequently used for
neutralization, leading to the generation of inorganic salts (mainly NaCl or Na₂SO₄) as
by-products^{9, 10}; on the other hand, the inorganic salts were also widely added as the
sedimentation agent to precipitate the dyes, thereby achieving rapid extraction of raw
dye products¹¹.

**References:**

9. Chen, P. et al. Performance of ceramic nanofiltration membrane for desalination of
dye solutions containing NaCl and Na₂SO₄. *Desalination* **404**, 102-111 (2017).

10. Yu, S., Gao, C., Su, H. & Liu, M. Nanofiltration used for desalination and
concentration in dye production. *Desalination* **140**, 97-100 (2001).

11. Yu, S. et al. Impacts of membrane properties on reactive dye removal from dye/salt
mixtures by asymmetric cellulose acetate and composite polyamide nanofiltration
membranes. *J. Membr. Sci.* **350**, 83-91 (2010).

2. Can the ultrafiltration and electrodialysis system used in this study effectively
separate dyes and inorganic salts in the presence of multiple inorganic salts, and what
are the effects of the types and concentrations of inorganic salts on the separation
process? What is the effect of the presence of other anions (such as SO_4^{2-}) on the
separation?

**Response to Reviewer comment No. 2:** We appreciate the reviewer for this comment.
In actual reactive dye synthesis, NaCl is the most common inorganic salt component,
as they are widely used as dyeing auxiliaries or generated as by products at various
stages of dye production and application. For instance, acid-base neutralization
reactions (generally, HCl is used to neutralize the base) during the dye synthesis
produce significant amounts of NaCl. Additionally, a large amount of NaCl would be
also added during the precipitation of the reactive dyes as raw dye product, which
reduce the purity of the dyes. In comparison, the usage and quantity of Na_2SO_4 are
relatively smaller, resulting in lower overall content in dye-containing wastewater.
Therefore, NaCl was chosen as a model salt to simulate a real dye-containing
wastewater in this study. As demonstrated in the manuscript, the tight ultrafiltration
membrane as anion conducting membrane can effectively realize the fractionation of
reactive dyes and NaCl during the electro-driven separation process.

On the other hand, as commented by the reviewers, sulfate ions may also exist during
the production of dyes, resulting in a high concentration of sulfate in the waste stream.
Indeed, the larger hydration radius and higher valence state of sulfate can potentially
affect the performance of the sub-4 nanometer porous membrane in electro-dialytic
process. In order to verify this hypothesis, we further tested the filtration performance
of the sub-4 nanometer porous membrane in the Na_2SO_4 solutions with different
salinities during the pressure-driven filtration process, as shown in Figure A3 below:

 **Figure A3. Pressure-driven filtration performance of the sub-4 nanometer**
 **porous membrane in pure Na₂SO₄ solution with different salinities. (A) Na₂SO₄**
 **rejection; (B) Permeability (Test at 4 bar and 25±1°C).**

 As demonstrated in **Figure A3**, the rejection of SO₄²⁻ for the sub-4 nanometer porous
 membrane is considerably high. For example, the rejection of SO₄²⁻ for the sub-4
 nanometer porous membrane is as high as 58.90% in 1.0 g·L⁻¹ Na₂SO₄ solution, which
 is much higher than that in 1.0 g·L⁻¹ NaCl solution (See Supplementary Fig. S2 below),
 since sulfate ions (SO₄²⁻) have larger hydration radius and higher valence state, yielding
 a stronger size exclusion and electrostatic repulsion effect by the sub-4 nanometer
 porous membrane with negative charge. As the concentration of Na₂SO₄ increased to
 14.6 g·L⁻¹, the rejection of Na₂SO₄ decreased to 7.79%, which is still much higher than
 that of NaCl.

 **Fig. S2. Pressure-driven filtration performance of the sub-4 nanometer porous**
 **membrane in pure NaCl solution with different salinities. (A) NaCl rejection; (B)**
 **Permeability.**

Furthermore, we also used the sub-4 nanometer porous membrane as anion conductive
 membrane for electro-driven desalination of Na₂SO₄ solution (~14.6 g·L⁻¹) (see **Figure**
 **A4** below) at the current intensity of 0.5 A. As indicated in **Figure A4A**, the sub-4
 nanometer porous membrane as anion conductive membrane shows an acceptable
 electro-driven SO₄²⁻ transfer capacity in the pure Na₂SO₄ solution with high
 concentrations. As the operation time prolongs, the concentration of SO₄²⁻ in the feed
 (diluate) decreases and the electrostatic repulsion against SO₄²⁻ was intensified,
 resulting in a slow SO₄²⁻ transfer at the low concentration level of SO₄²⁻, which can be
 reflected by the slow decay in feed conductivity. Compared to the case in pure NaCl
 solution as feed, the sub-4 nanometer porous membrane as anion conductive membrane
 has a poorer anion transfer capacity in the pure Na₂SO₄ solution. The sub-4 nanometer
 porous membrane requires a longer time for desalination of Na₂SO₄ solution (436-min
 operation) than that of NaCl solution (216-min operation) (**Figure A4A**). The electro-
 driven desalination efficiency of pure NaCl and Na₂SO₄ solution (~14.6 g·L⁻¹) is
 98.29% and 97.50% by the sub-4 nanometer porous membrane, respectively (**Figure**
 **A4B**). Furthermore, the salt transfer rate of the sub-4 nanometer porous membrane in
 the pure Na₂SO₄ solution (1.95 g·L⁻¹·h⁻¹) is much lower than that in the pure NaCl
 solution (3.99 g·L⁻¹·h⁻¹) (see (**Figure A4C** and **Table A1** below).

**Figure A4. Electrodialytic filtration performance of the sub-4 nanometer porous**
 **membrane (MWCO of 2292 Da) as anion conductive membrane in pure NaCl**
 **and Na₂SO₄ solution (~14.6 g·L⁻¹) as feed (diluate) at current intensity of 0.5 A.**
 **(A) Evolution of conductivity in the diluate solution, (B) NaCl/Na₂SO₄**
 **concentration in the diluate and desalination efficiency, (C) NaCl and Na₂SO₄**
 **transfer rate.**

 **Table A1** Electro-driven desalination performance of the sub-4 nanometer porous
 membrane (MWCO of 2292 Da) as a role of anion conductive membrane in pure
 NaCl and Na₂SO₄ solutions (salt concentration of ~14.6 g·L⁻¹)

Salt	NaCl	Na ₂ SO ₄
Operation time for desalination, min	216 min	436 min
Desalination efficiency, %	98.29%	97.50%
Salt transfer rate	3.99 g·L ⁻¹ ·h ⁻¹	1.95 g·L ⁻¹ ·h ⁻¹

 Therefore, the fast permeation of SO₄²⁻ ions through sub-4 nanometer porous membrane
 as anion conductive membrane is still challenging for both pressure-driven or electro-
 driven nanofiltration membranes, due to intrinsically strong electrostatic repulsion
 between SO₄²⁻ (higher valent than Cl⁻) and the nanoporous membranes with negative
 charges (most of commercial and fabricated nanoporous membranes carry negative
 charges). Therefore, it is very important to tailor the surface properties (surface charge
 and pore size) for both the pressure-driven and electro-driven nanoporous membranes.
 According to the reviewer's comment, we are inspired that the electro-neutral
 membranes (with zeta potential of ~0 mV) are preferred to be a feasible solution to
 enhance the transfer of sulfate through significantly reducing the electrostatic repulsion
 between sulfate and electro-nanofiltration membranes. In future, we will extend our
 work in fabrication of advanced electro-neutral nanoporous membranes for
 electrodialytic separation in high-valent salts (e.g., sulfate). Moreover, the interfacial
 polymerization and surface coating methods for membrane fabrication will be firstly
 taken into consideration to reduce the thickness of the membrane selective layer (to be

the thickness of 50~100 nm) for shortening the anion transfer pathway and thus
enhancing the anion transfer. In addition, the pore size, surface charge, and surface
hydrophilicity of the membrane will be precisely regulated as well to enhance the
transfer of anion (e.g., sulfate) BUT improve dye rejection. We thank the reviewer again
to inspire us. We believe that following the suggestion of the reviewer, more exciting
outcomes will be achieved to address the technical challenge in electro-driven transfer
of high-valent salts (e.g., sulfate).

3. Only one dye (RB5) was used in the study, but did the other dyes have a similar
effect, how much of a role did the aggregation effect of the dye play in the separation
process, and what was the separation effect for the dyes with a less aggregation effect?

**Response to Reviewer comment No. 3:** We thank the reviewer very much for this
comment! This comment inspires us a lot. In order to test the practicability of the tight
ultrafiltration process using sub-4 nm porous membrane for dye rejection, we use
different dyes (wide molecular weights ranging from 626.5 to 1338.1 g·mol⁻¹, see the
detailed dye information in the section of “**Materials and chemicals**”) for filtration
experiment (we place the separation results as Supplementary Fig. S4, see below as
well):

**Fig. S4. Pressure-driven separation performance of the sub-4 nanometer porous**
**membrane in different reactive dye solutions (dye concentration of 1.0 g·L⁻¹).**

As indicated in Supplementary Fig. S4, the reviewer can see that apart from reactive
black 5 dye (molecular weight: $991.8 \text{ g}\cdot\text{mol}^{-1}$), the sub-4 nanometer porous membrane
in this work has the similarly high rejection efficiency ($> 99.4\%$) for another 7 reactive
dyes (wide molecular weights ranging from 626.5 to $1338.1 \text{ g}\cdot\text{mol}^{-1}$), indicating that
the impressive orientation of the sub-4 nanometer porous membrane for reactive dye
retention, due to the fact that other reactive dyes with different molecular weights (even
small molecular weights of $626.5 \text{ g}\cdot\text{mol}^{-1}$) have the similar aggregation effect with
reactive black 5. Specifically, the reactive blue 19 dye (molecular weight of 626.5
$\text{g}\cdot\text{mol}^{-1}$) has a much smaller size than the pore size of the sub-4 nanometer porous
membrane (2292 Da MWCO), which can easily pass through the membrane. Therefore,
the aggregation effect of the dyes (especially for dyes with small sizes) has the extreme
importance on their high rejection by the sub-4 nm porous membrane. Such results also
demonstrate that the sub-4 nanometer tight ultrafiltration membrane can be a useful and
powerful membrane for dye/salt separation at both pressure-driven and electro-driven
modes, showing a great potential as alternative of the nanofiltration membrane for
dye/salt separation.

Generally, it is not easy to quantify the aggregation effect (aggregation degree) of the
dyes. Very limited literature has shown the aggregation degree of the dyes in the
solutions. Indeed, we thank the reviewer very much for inspiring us. It is important and
useful to reveal the aggregation behaviors of the dyes for the membrane researchers,
which is important to illustrate the filtration performance of the tested membrane and
offer a guideline for design of high-performance membranes in dye separation. In our
group, we will do more work to reveal the aggregation behavior of the dyes in future to
offer more exciting outcomes to the potential readers. We thank the reviewer again!

4. In Figure 6D, chloride ions diffuse from the anode to the cathode under the action of
an electric field, why do the dye molecules with the same negative charge not
accumulate on the surface of the film with the action of the electric field? Because
whether it is a simple ultrafiltration or ultrafiltration + electro dialysis system, there is
charge repulsion on the negatively charged dye and negatively charged membrane
surface, and the pressure of the ultrafiltration process alone will pollute the membrane,
why will the electro dialysis process not pollute the membrane under the action of the

electric field force (the negatively charged dye is forced to the membrane surface and
migrates to the membrane surface)?

**Response to Reviewer comment No. 4:** We thank the reviewer for this insightful
comment. Indeed, under the applied electric field, chloride ions (Cl^-) migrate from the
cathode to the anode through the nanochannels of the tight ultrafiltration membrane, as
depicted in Fig. 6D. For clarity, this figure illustrates only the anion transfer pathway,
omitting the cation (Na^+) movement through the cation exchange membrane.
Consequently, the membrane's inherent size exclusion properties effectively retain the
dye molecules. In contrast, conventional pressure-driven filtration with the tight
ultrafiltration membrane typically leads to the formation of a compact cake layer on the
membrane surface, which at high pressures (e.g., 4 bar) can block the membrane pores.
This effect is evidenced by a 54.2% reduction in membrane flux (see Fig. 2 below)
during the filtration of the dye/salt mixture. However, in the electro-driven filtration
system (i.e., ultrafiltration-based electrodialysis), the electric field not only facilitates
the rapid transfer of Cl^- ions through the membrane but also ensures that the dye
molecules, which exhibit extremely low transfer velocities (as shown in the molecular
dynamics simulation in Fig. 5D), remain largely stationary. Moreover, the negatively
charged surface of the tight ultrafiltration membrane further repels the similarly charged
dye molecules via electrostatic interactions. Although the electric field may drive some
dye molecules toward the membrane surface, any resulting loose dye cake layer is easily
removed by a simple water flush. As a result, when employed as an anion-conducting
membrane, the tight ultrafiltration membrane exhibits excellent antifouling
performance over eight electro-dialytic filtration cycles of the dye/salt mixture, with
water flushing between cycles effectively eliminating any dye accumulation.

**Fig. 2 Filtration performance of the sub-4 nanometer porous membrane in**
**reactive black 5/NaCl mixture solution with different salinities. (A) Rejection of**
**solute (reactive black 5 and NaCl) and permeability.**

5. Although the ultrafiltration process alone will be subject to more serious membrane
fouling, the whole filtration system is relatively simple, and the membrane fouling can
reduce the harm of membrane fouling through regular membrane cleaning. However,
the ultrafiltration + electro dialysis system is more complex (requires an electric field,
requires an additional cation exchange membrane), and in terms of overall cost and
economic benefits, is there a better overall effect in this study?

**Response to Reviewer comment No. 5:** We thank the reviewer for noting the
simplicity of pressure-driven ultrafiltration and the potential for mitigating fouling via
membrane cleaning. However, we would like to emphasize that this approach has
notable limitations for desalinating dye/salt mixtures. First of all, the flux of the
ultrafiltration membrane will be reduced (reduction in flux by 16.9% in this study)
during the filtration of dye solution, due to the formation of compact dye cake layer for
significantly enhanced hydraulic resistance (see the updated supplementary Figure S3
below). Specifically, the flux of the ultrafiltration membrane will be further reduced to
large extent (reduction in flux by 54.2%, compared to pure water) during the filtration
of dye/salt mixture solution with elevated salinity (see Figure 2A below, marked in
purple), which can remarkably deteriorate the separation efficacy of the tight
ultrafiltration membrane during the pressure-driven filtration of dye/salt mixture
solution.

**Fig. S3. Pressure-driven filtration performance of the sub-4 nanometer porous**
 **membrane in pure reactive black 5 solutions with different concentrations. (A)**
 **Dye rejection; (B) Permeability.**

 **Fig. 2 Filtration performance of the sub-4 nanometer porous membrane in**
 **reactive black 5/NaCl mixture solution with different salinities. (A) Rejection of**
 **solute (reactive black 5 and NaCl) and permeability.**

On the other hand, the ultrafiltration-based diafiltration process (i.e., continuous
 addition of pure water to remove the salt from dye/salt mixture solution) is inevitably
 applied to effective fractionation of dye and salt for dye purification (see the updated
 supplementary figure S13 below, which represents the ultrafiltration-based constant-
 volume diafiltration process for dye/salt separation) (X. Wang, C. Zhang, P. Ouyang,
 *J. Membr. Sci.*, 2002, 204, 271-281). The ultrafiltration-based diafiltration process
 requires a large consumption of pure water to purify the dye. In the case of the study,
 we need a dia-volume number (η) of 4.6 (ratio between volume of the pure water added
 and volume of the feed, Figure 3A) to desalinate the dye/salt mixture solution for
 separation of dye and salt (salt removal efficiency of 98.07%). Furthermore, the salt in
 the permeate will be diluted by the added pure water at a low concentration level, which
 fails to be recycled.

**Figure S13 Schematic of the pressure-driven constant-volume diafiltration using**
 **the sub-4 nanometer porous membrane in the reactive black 5 /NaCl mixture**
 **solution for fractionation of dye/NaCl mixture.**

**Fig. 3 Pressure-driven constant-volume diafiltration using the sub-4 nanometer**
 **porous membrane in the reactive black 5 /NaCl mixture solution for fractionation**
 **of dye and NaCl. (A) Content of reactive black 5 and NaCl at different diavolumes;**
 **(B) Loss of reactive black 5 and desalination efficiency.**

Alternatively, electrodialysis unlocks the potential to desalinate high-salinity dye-
 containing liquor, which enables the directed transfer of anion and cation through anion
 exchange membranes and cation exchange membranes under a direct current field.
 However, dyes with negative charges would preferentially accumulate onto the surface
 or pore structure of anion exchange membranes (with positive charges) via the
 electrostatic attraction effect, inducing severe membrane fouling and deteriorating the
 desalination efficacy.

Therefore, in this study, we integrated the technical merits of (tight) ultrafiltration
 membrane and conventional electro dialysis process to construct the electro-driven
 filtration system using tight ultrafiltration membrane as anion conducting membrane
 for effective anion transfer and desalination of dye/salt mixture. As indicated in Figure
 6 (see below as well), the tight ultrafiltration membrane as anion conducting membrane
 shows the consistently effective and one-step fractionation efficacy for dye and salt
 (desalination efficiency of; dye recovery of 98.15% desalination efficiency and 99.66%
 dye recovery) (See Supplementary Table S1 below) with impressive antifouling
 performance, which markedly outperforms the commercial anion exchange membrane.

Fig. 6 Eight-cycle electro dialytic filtration using the sub-4 nanometer porous membrane as anion conducting membrane for fractionation of dye and NaCl from the reactive black 5/NaCl mixture solution. (A) Evolution of conductivity in the concentrate and diluate, (B) NaCl content in the diluate and desalination efficiency, (C) Dye content in the concentrate and recovery rate, (D) Illustration of anion (Cl⁻ and reactive black 5) transfer of the sub-4 nanometer porous membrane for electro dialytic fractionation of the reactive black 5/NaCl mixture solution.

**Table S1.** Performance of the pressure-driven diafiltration and electro-driven filtration
 process for fractionation of reactive dye and NaCl using the sub-4 nanometer porous
 membrane

Mode	NaCl content in feed, g·L ⁻¹	Desalination efficiency, %	Recovery of reactive dye, %
Pressure-driven diafiltration	0.28	98.07	98.29
Electro-driven filtration	0.27	98.15	99.66

In this study, the electro-driven filtration system using tight ultrafiltration membrane as
 anion conducting membrane not only has the intrinsic advantages of both ultrafiltration
 and conventional electro dialysis processes, but also avoids their technical defects
 (mainly membrane fouling). The main advantages of the electro-driven filtration
 process are summarized below: (1) the electro-driven filtration process can achieve one-
 step fractionation of dye and salt under the electric field. And the separated salt can be
 recovered at a higher concentration level in the concentrate chamber. However, the
 pressure-driven filtration process needs a constant-volume diafiltration procedure
 (which consumes large volume of pure water for desalination) for fractionation of dye
 and salt, which can allow for the loss of dye with the permeate as well (electro-driven
 filtration process can obtain high dye recovery). The reviewer can see the results in
 Supplementary Table S1 above; (2) the electro-driven filtration process using tight
 ultrafiltration membrane only need a membrane stack and electrical power supply as
 driving force. The system is also simple as well. However, the pressure-driven filtration
 process needs a pressure vessel and a pump which needs to supply the high pressure for
 separation of dye and salt; (3) compared to the conventional electro dialysis process, the
 electro-driven filtration process using tight ultrafiltration membrane as anion
 conducting membrane can achieve a higher fractionation efficiency for dyes and salt
 with excellent antifouling performance (see Figure 6 and updated Supplementary
 Figure S12 below). The tight ultrafiltration membrane as anion conducting membrane
 has a lower cost for membrane fabrication and higher separation performance for
 organics/salt mixture solution, compared to the commercial anion exchange membrane.

**Fig. S12. Three-cycle electrodialytic filtration using commercial anion exchange**
 **membrane for fractionation of dye and NaCl from the reactive black 5/NaCl**
 **mixture solution. (A) Evolution of conductivity in the concentrate and diluate**
 **solutions, (B) NaCl content in the diluate and desalination efficiency, (C) Illustration**
 **of anion (Cl⁻ and reactive black 5) transfer of commercial anion exchange membrane**
 **during electrodialytic fractionation of the reactive black 5/NaCl mixture solution, (D)**
 **Specific areal electric resistance of commercial anion exchange membrane before and**
 **after fouling.**

Therefore, the electro-driven filtration process using tight ultrafiltration membrane as
 anion conducting membrane in this study has the distinct advantages and technical
 merits in fractionation of organics/salt mixture solution, as demonstrated in the
 manuscript. In this study, we don't compare the electro-driven filtration with other
 membrane processes. The aim and motivation of our study is to raise a new concept and
 solution to demonstrate the practicability of tight ultrafiltration membrane as anion
 conducting membrane in effective electro-driven fractionation for dye and salt, which
 may help the potential readers to extend the application of tight ultrafiltration

membranes at an electro-driven process, in addition to pressure-driven process. If the
proposed electro-driven filtration process of the tight ultrafiltration membrane as anion
conducting membrane in this study can be proved to be useful in future by the potential
readers and membrane researchers, it will be a great honor and proudness for us, since
we make a contribution to the membrane society. In addition, the tight ultrafiltration
membrane has a low cost for membrane fabrication, facile fabrication process (i.e., non-
solvent phase inversion) and higher performance for organics/salt mixture solution.
Therefore, our study can provide a new clue to the membrane researcher for large-scale
fabrication of high-performance anion conducting membrane as well.

We thank the reviewer again for inspiring us. In future, we will extend our work in
high-performance membrane design and fabrication for versatile applications. And the
economic benefits of the electro-driven filtration process using tight ultrafiltration
membrane as anion conducting membrane will be systematically evaluated as well.

6. Figures 5E and 5F mentioned in the discussion do not exist in the figures.

**Response to Reviewer comment No. 6:** We apologize for the typo. Actually, they are
figures 5c and 5d, rather than figure 5e and 5f. Now we revise that in the Discussion
section of the manuscript to avoid the misunderstanding to the reviewer and the
potential readers. The reviewer can see that below as well:

**Discussion**

The tested sub-4 nanometer membrane established an enhanced anion electro-dialytic
transport via its intrinsically nanoporous structure. Molecular dynamic simulations of
anion transport were further conducted to illustrate the ion transfer mechanism of the
sub-4 nanometer porous membrane as anion conducting membrane under a direct
current field (**Fig. 5C, Fig. 5D** and **Supplementary Movies S5**).

7. Figure S10 B is incorrect and inconsistent with the description "commercial anion
exchange membrane declined from 92.87% to 84.98% after the three-cycle
electro-dialytic filtration operation".

**Response to Reviewer comment No. 7:** We thank the reviewer very much for this
comment! We deeply apologised that we made a mistake in the previous Figure S10B.
We place the mistaken direction of the arrows for data illustration in the previous Figure

S10B. Now we updated the Figure S10B now through revising the direction of arrow
for data illustration to make it consistent with the statement of "commercial anion
exchange membrane declined from 92.87% to 84.98% after the three-cycle
electro dialytic filtration operation". The reviewer can see the updated supplementary
Fig. S12B (the left figure shows the wrong direct of the arrows for data illustration; the
right figure below is the updated figure, which shows the correct direct of the arrows
for data illustration):

Response to the Reviewers

The authors appreciate the Editor and Reviewers for their constructive comments and suggestions on our revised manuscript. The followings are our point-by-point responses to Reviewers' comments.

Reviewer #1 (Remarks to the Author):

The authors have satisfactorily addressed the reviewers' concerns and conducted sufficient supplementary research and revisions to the manuscript. Therefore, I recommend acceptance for publication.

Response to Reviewer comment: We thank the reviewer for his/her positive assessment and constructive comments.

Reviewer #2 (Remarks to the Author):

The paper is well revised and concerns are replied properly.

Response to Reviewer comment: We thank the reviewer for his/her positive assessment of our manuscript and constructive comments.